# Provably Auditing Ordinary Least Squares in Low Dimensions

**Ankur Moitra & Dhruv Rohatgi**
Massachusetts Institute of Technology
{moitra, drohatgi}@mit.edu

## Abstract

Auditing the stability of a machine learning model to small changes in the training procedure is critical for engendering trust in practical applications. For example, a model should not be overly sensitive to removing a small fraction of its training data. However, algorithmically validating this property seems computationally challenging, even for the simplest of models: Ordinary Least Squares (OLS) linear regression. Concretely, recent work defines the stability of a regression as the minimum number of samples that need to be removed so that rerunning the analysis overturns the conclusion (Broderick et al., 2020), specifically meaning that the sign of a particular coefficient of the OLS regressor changes. But the only known approach for estimating this metric, besides the obvious exponential-time algorithm, is a greedy heuristic that may produce severe overestimates and therefore cannot certify stability. We show that stability can be efficiently certified in the *low-dimensional* regime: when the number of covariates is a constant but the number of samples is large, there are polynomial-time algorithms for estimating (a fractional version of) stability, with provable approximation guarantees. Applying our algorithms to the Boston Housing dataset, we exhibit regression analyses where our estimator outperforms the greedy heuristic, and can successfully certify stability even in the regime where a constant fraction of the samples are dropped.

## 1 Introduction

A key facet of interpretability of machine learning models is understanding how different subsets of the training data influence the learned model and its predictions. Computing the influences of individual training points has been shown to be a useful tool for enhancing trust in the model (Zhou et al., 2019), for tracing the origins of model bias (Brunet et al., 2019), and for identifying mislabelled training data and other model debugging (Koh & Liang, 2017). Modelling the influence of groups of training points has applications to measuring fairness (Chen et al., 2018), vulnerability to contamination of multi-source training data (Hayes & Ohrimenko, 2018), and (most relevant to this paper) identification of unstable predictions (Ilyas et al., 2022) and models (Broderick et al., 2020). In a high-stakes machine learning application, it would likely be alarming if some data points were so influential that the removal of, say, $1\%$ of the training data dramatically changed the model. An ideal, trustworthy machine learning pipeline therefore should include validation that this does not happen.

But the obvious algorithm for checking if a model trained on $n$ data points exhibits this instability would require computing the group influences of $\binom{n}{n/100}$ different subsets of the data, which is computationally infeasible even for fairly small $n$. Instead, current methods for estimating the stability of a model simply use the first-order approximation of group influence: namely, the sum of individual influences of data points in the group. With this approximation, vulnerability of a model to dropping $\alpha n$ data points is heuristically estimated by dropping the $\alpha n$ most individually influential data points (Broderick et al., 2020; Ilyas et al., 2022).

This heuristic can be thought of as using "local" stability as a proxy for "global" stability, and it has found substantial anecdotal success in diagnosing *unstable* models. Unfortunately, for correlated groups of data points, the first-order approximation of the group influence is often an underestimate (Koh et al., 2019), so large local stability does not actually certify that a model is provably *stable* to

removing small subsets of data. In fact, stability certification is a challenging and open problem even in the simplest of models: linear regression via Ordinary Least Squares (OLS).

Concretely, given a regression dataset, a natural metric for the stability of the OLS regressor is the minimum number of data points that need to be removed from the dataset to flip the sign of a particular coefficient of the regressor (e.g., in causal inference, the coefficient measuring the treatment effect). Recent work has used the local stability heuristic to diagnose unstable OLS regressions in several prominent economics studies (Broderick et al., 2020), identifying examples where even a statistically significant conclusion can be overturned by removing less than $1\%$ of the data points. However, the converse question of validating stable conclusions remains unaddressed:

*Given a regression dataset, can we efficiently certify non-trivial lower bounds on the stability of the OLS regressor?*

Our work takes steps towards addressing this question, via the following contributions:

- We introduce a natural fractional relaxation of the above notion of OLS stability, where we allow removing fractions of data points, and seek to minimize the total removed weight. We call this *finite-sample stability*, and henceforth refer to the prior notion as "integral" stability.

- We develop approximation algorithms for the finite-sample stability, with **(a)** provable guarantees under reasonable anti-concentration assumptions on the dataset, and **(b)** running time polynomial in the size of the dataset, so long as the dimension of the data is a constant (in contrast, the naive algorithm is exponential in the size of the dataset). Moreover, we prove that (at least for exact algorithms) exponential dependence of the running time on the dimension is unavoidable under standard complexity assumptions.

- We use modifications of our algorithms to compute *assumption-free upper and lower bounds* on the finite-sample stability of several simple synthetic and real datasets, achieving tighter upper bounds than prior work and the first non-trivial lower bounds, i.e. certifications that the OLS regressor is stable.

**Why define stability this way?** The definition of integral stability was introduced in (Broderick et al., 2020), along with several variants (e.g. smallest perturbation which causes the first coordinate to lose significance). We choose the definition based on the sign of the first coordinate, because it has clear practical interpretation—does the first covariate positively or negatively affect the response?—which does not depend on choice of additional parameters such as significance level.

We study the fractional relaxation so that the stability is defined by a continuous optimization problem. Note that certifying a lower bound on fractional stability immediately certifies a lower bound on the integral stability; we will see later (Remark 3.1) that a near-converse holds in low dimensions.

**Why is low-dimensional regression important?** Given that much of machine learning happens in high-dimensional settings, where the number of covariates can even be larger than the number of datapoints, it is natural to wonder why low-dimensional settings are still important. First, in application areas such as econometrics, linear regressions with as few as two to four covariates are very common (Britto et al., 2022; Bianchi & Bigio, 2022; Hopenhayn et al., 2022), often serving as proofs-of-concept for more complex models. Second, even in settings where the number of covariates is larger, it is often the expectation that few covariates are relevant. In such applications, analysis often consists of a variable selection step followed by regression on a much-reduced set of covariates (Cai & Wang, 2011). In all these settings, understanding the stability of an estimator is important, and our work gives some of the first provable guarantees that avoid making strong distributional assumptions. Moreover our lower bounds show that certifying stability of truly high-dimensional models, even linear ones, is intractable.

## 1.1 FORMAL PROBLEM STATEMENT

We are given a deterministic and arbitrary set of $n$ samples $(X_i, y_i)_{i=1}^n$, where each $X_i$ is a vector of $d$ real-valued *covariates*, and each $y_i$ is a real-valued *response*. We are interested in a single coefficient of the OLS regressor (without loss of generality, the first coordinate): in an application, the first covariate may be the treatment and the rest may be controls. The sign of this coefficient

is important because it estimates whether the treatment has a positive or negative effect. Thus, we want to determine if it can be changed by dropping a few samples from the regression. Formally, we consider the fractional relaxation, where we allow dropping fractions of samples:

**Definition 1.1.** Fix $(X_i, y_i)_{i=1}^n$ with $X_1, \ldots, X_n \in \mathbb{R}^d$ and $y_1, \ldots, y_n \in \mathbb{R}$. For any $w \in [0, 1]^n$, the *weight-$w$ OLS solution set* of $(X_i, y_i)_{i=1}^n$ is

$$\mathsf{OLS}(X, y, w) := \arg\min_{\beta \in \mathbb{R}^d} \frac{1}{n} \sum_{i=1}^n w_i(\langle X_i, \beta \rangle - y_i)^2.$$

The *finite-sample stability* of $(X_i, y_i)_{i=1}^n$ is

$$\mathrm{Stability}(X, y) := \inf_{w \in [0,1]^n, \beta \in \mathbb{R}^d} \{n - \|w\|_1 : \beta_1 = 0 \text{ and } \beta \in \mathsf{OLS}(X, y, w)\}.$$

This is the minimum number of samples (in a fractional sense) which need to be removed to zero out the first coordinate of the OLS regressor. If the OLS solution set contains multiple regressors, then it suffices if any regressor $\beta$ in the solution set has $\beta_1 = 0$. Our algorithmic goal is to compute $\mathrm{Stability}(X, y)$, or at least to approximate $\mathrm{Stability}(X, y)$ up to an additive $\epsilon n$ error.

## 1.2 RESULTS

By brute-force search, the (integral) stability can be computed in time $2^n \cdot \mathrm{poly}(n)$. However, because the complexity is exponential in the number of samples, it is computationally infeasible even when the dimension $d$ of the data is low, which is a common situation in many scientific applications. Similarly, the fractional stability (Definition 1.1) is the solution to a non-convex optimization problem in more than $n$ variables, which seems no simpler. Can we still hope for a *polynomial-time* algorithm in constant dimensions? We show that the answer is yes.

**Theorem 1.2.** *There is an $n^{O(d^3)}$-time algorithm which, given $n$ arbitrary samples $(X_i, y_i)_{i=1}^n$ with $X_1, \ldots, X_n \in \mathbb{R}^d$ and $y_1, \ldots, y_n \in \mathbb{R}$, and given $k \geq 0$, decides whether $\mathrm{Stability}(X, y) \leq k$.*

We also show that the exponential dependence on dimension $d$ is necessary under standard complexity assumptions:

**Theorem 1.3.** *Under the Exponential Time Hypothesis, there is no $n^{o(d)}$-time algorithm which, given $(X_i, y_i)_{i=1}^n$ and $k \geq 0$, decides whether $\mathrm{Stability}(X, y) \leq k$.*

This theorem in particular rules out fixed-parameter tractability, i.e. algorithms with time complexity $f(d) \cdot \mathrm{poly}(n)$. However, it only applies to *exact* algorithms. In practice, it is unlikely to matter whether $\mathrm{Stability}(X, y) = 0.01n$ or $\mathrm{Stability}(X, y) = 0.02n$; in both cases, the conclusion is sensitive to dropping a very small fraction of the data. This motivates our next two algorithmic results on $\epsilon n$-additive approximation of the stability (where we think of $\epsilon > 0$ as a constant). First, we make a mild anti-concentration assumption, under which the stability can $\epsilon n$-approximated in time roughly $n^{d+O(1)}$. While still not fixed-parameter tractable, this algorithm can now be run on moderate sized problems in low dimensions, unlike the algorithm in Theorem 1.2.

**Assumption A.** *Let $\epsilon, \delta > 0$. We say that samples $(X_i, y_i)_{i=1}^n$ satisfy $(\epsilon, \delta)$-anti-concentration if for every $\beta \in \mathbb{R}^d$, it holds that*

$$\left| \left\{ i \in [n] : |\langle X_i, \beta \rangle - y_i| < \frac{\delta}{\sqrt{n}} \left\| X\beta^{(0)} - y \right\|_2 \right\} \right| \leq \epsilon n,$$

*where $X : n \times d$ is the matrix with rows $X_1, \ldots, X_n$, and $\beta^{(0)} \in \mathsf{OLS}(X, y, \mathbb{1})$ is any unweighted OLS regressor of $y$ against $X$.*

See Appendix F.1 for discussion of when Assumption A holds. Under this assumption, we present an $O(\epsilon n)$-approximation algorithm:

**Theorem 1.4.** *For any $\epsilon, \delta, \eta > 0$, there is an algorithm* PARTITIONANDAPPROX *with time complexity*

$$\left( n + \frac{Cd}{\epsilon^2} \log \frac{1}{\delta} \log \frac{1}{\epsilon \eta} \right)^{d+O(1)}$$

*which, given $\epsilon$, $\delta$, $\eta$, and samples $(X_i, y_i)_{i=1}^n$ satisfying $(\epsilon, \delta)$-anti-concentration, returns an estimate $\hat{S}$ such that with probability at least $1 - \eta$,*

$$|\hat{S} - \text{Stability}(X, y)| \leq 12\epsilon n + 1.$$

In fact, PARTITIONANDAPPROX also can detect when Assumption A fails (see Theorem D.6 for a precise statement), so it can be used to compute unconditional lower bounds on stability with high probability (where the lower bound is provably tight if the data satisfies anti-concentration). Moreover, as discussed in Appendix F.1, the required anti-concentration is very mild. If $\epsilon, \eta > 0$ are constants, the algorithm has time complexity $n^{d+O(1)}$, so long as the samples satisfy $(\epsilon, \exp(-\Omega(n)))$-anti-concentration. This is true for arbitrary *smoothed* data. Finally, unlike the exact algorithm, PARTITIONANDAPPROX avoids heavy algorithmic machinery; it only requires solving linear programs.

**Fixed-parameter tractability?**     Our final result is that $\epsilon n$-approximation of the stability is in fact fixed-parameter tractable, under a stronger anti-concentration assumption.

**Assumption B.** *Let $\epsilon, \delta > 0$. We say that samples $(X_i, y_i)_{i=1}^n$ satisfy $(\epsilon, \delta)$-strong anti-concentration if for every $\beta \in \mathbb{R}^{d+1}$, it holds that*

$$\left| \left\{ i \in [n] : |\langle \overline{X}_i, \beta \rangle| < \frac{\delta}{\sqrt{n}} \left\| \overline{X}\beta \right\|_2 \right\} \right| \leq \epsilon n$$

*where $\overline{X} : n \times (d+1)$ is the matrix with columns $(X^T)_1, \ldots, (X^T)_d, y$.*

This assumption holds with constant $\epsilon, \delta > 0$ under certain distributional assumptions on $(X_i, y_i)_{i=1}^n$, e.g. centered Gaussian mixtures with uniformly bounded condition number (Appendix F.2).

**Theorem 1.5.** *For any $\epsilon, \delta > 0$, there is a $(\sqrt{d}/(\epsilon\delta^2))^d \cdot \text{poly}(n)$-time algorithm NETAPPROX which, given $\epsilon, \delta$, and samples $(X_i, y_i)_{i=1}^n$ satisfying $(\epsilon, \delta)$-strong anti-concentration, returns an estimate $\hat{S}$ satisfying $\text{Stability}(X, y) \leq \hat{S} \leq \text{Stability}(X, y) + 3\epsilon n + 1$. Moreover, $\text{Stability}(X, y) \leq \hat{S}$ holds for arbitrary $(X_i, y_i)_{i=1}^n$.*

**Extensions.**     Another model, frequently used in causal inference and econometrics, is *instrumental variables (IV) linear regression*. When the noise $\eta$ in a hypothesized causal relationship $y = \langle X, \beta^* \rangle + \eta$ is believed to be endogenous (i.e. correlated with $X$), a common approach (Sargan, 1958; Angrist et al., 1996; Card, 2001) is to find a $p$-dimensional variable $Z$ (the *instrument*) for which domain knowledge suggests that $\mathbb{E}[\eta | Z] = 0$. Positing that $\beta^*$ is identified by the moment condition $\mathbb{E}[Z(y - \langle X, \beta \rangle)] = 0$, the IV estimator set given samples $(X_i, y_i, Z_i)_{i=1}^n$ is then

$$\text{IV}(X, y, Z) = \{\beta \in \mathbb{R}^d : Z^T(w \star (X\beta - y)) = 0\}$$

where $a \star b$ denotes elementwise product, and $Z : n \times p$ and $X : n \times d$ are the matrices of instruments and covariates respectively. Stability can be defined as in Definition 1.1. Although for simplicity we state all of our results for OLS (i.e. the special case $Z = X$), it can be seen that Theorem 1.2 and Theorem 1.5 both extend directly to the IV regression setting. See Appendix G for further discussion.

**Experiments.**     We implement modifications of NETAPPROX and PARTITIONANDAPPROX which give *unconditional*, *exact* upper and lower bounds on stability, respectively. We use these algorithms to obtain tight data-dependent bounds on stability of isotropic Gaussian datasets for a broad range of signal-to-noise ratios, and we demonstrate heterogeneous synthetic datasets where our algorithms' upper bounds are an order of magnitude better than upper bounds obtained by the prior heuristic. On the Boston Housing dataset (Harrison Jr & Rubinfeld, 1978), we regress house values against all pairs of features. For the majority of these regressions, we bound the stability within a factor of two. On the one hand, we detect many sensitive conclusions (including some which the greedy heuristic claims are stable); on the other hand, we certify that some conclusions are stable to dropping as much as half the dataset.

## 1.3   ORGANIZATION

In Section 2 we review related work. In Section 3 we collect notation and formulas that will be useful later. In Section 4 we sketch the intuition behind our algorithmic results. Section 5 covers our experiments. In Appendices B, C, D, and E we prove Theorems 1.2, 1.3, 1.4, and 1.5 respectively.

## 2 RELATED WORK

There is a rich literature on topics related to finite-sample stability, including sensitivity analysis and robustness to distribution shift and contamination. Due to space constraints, here we only discuss the works most relevant to ours, and we postpone broader discussion to Appendix A.

Most directly related is the prior work on heuristics for the (integral) stability (Broderick et al., 2020; Kuschnig et al., 2021). The heuristic given by Broderick et al. (2020) (to approximate the most-influential $k$ samples) is simply the *local* approximation: compute the local influence of each sample at $w = \mathbb{1}$, sort the samples from largest to smallest influence, and output the top $k$ samples. Subsequent work (Kuschnig et al., 2021) refines this heuristic by recomputing the influences after removing each sample, which alleviates issues such as masking (Chatterjee & Hadi, 1986). But this is still just a greedy heuristic, and it may fail when samples are jointly but not individually influential. Except under the strong assumption that the sample covariance remains nearly constant when we remove any $\epsilon n$ samples (see Theorem 1 in Broderick et al. (2020), which relies on Condition 1 in Giordano et al. (2019)), the local influence approach can upper bound the finite-sample stability but cannot provably lower bound it. In fact, in Section 5 we provide examples where the greedy heuristic of Kuschnig et al. (2021) is very inaccurate due to instability in the sample covariance.

Closely related to finite-sample stability, the $s$-value (Gupta & Rothenhäusler, 2021) is the minimum Kullback-Leibler divergence $D(P||P_0)$ over all distributions $P$ for which the conclusion is null, where $P_0$ is the empirical distribution of the samples. Unfortunately, while the $s$-value is an interesting and well-motivated metric, computing the $s$-value for OLS estimation appears to be computationally intractable, and the algorithms given by Gupta & Rothenhäusler (2021) lack provable guarantees.

## 3 PRELIMINARIES

For vectors $u, v \in \mathbb{R}^m$, we let $u \star v$ denote the elementwise product $(u \star v)_i = u_i v_i$. Throughout the paper, we will frequently use the closed-form expression for the (weighted) OLS solution set

$$\mathsf{OLS}(X, y, w) = \{\beta \in \mathbb{R}^d : X^T(w \star (X\beta - y)) = 0\}$$

where $X : n \times d$ is the matrix with rows $X_1, \ldots, X_n$. In particular, setting $\lambda = \beta_{2:d}$, this means that the finite-sample stability can be rewritten as

$$\text{Stability}(X, y) = \inf_{w \in [0,1]^n, \lambda \in \mathbb{R}^{d-1}} \{n - \|w\|_1 : X^T(w \star (\tilde{X}\lambda - y)) = 0\} \tag{1}$$

where (here and throughout the paper) $\tilde{X} : n \times (d-1)$ is the matrix with columns $(X^T)_2, \ldots, (X^T)_d$.

**Remark 3.1.** As previously mentioned, the finite-sample stability always lower bounds the integral stability (the minimum number of samples that need to be removed to make the first coordinate of the regressor change sign), by continuity of the OLS solution set in $w$. Additionally, it can be seen from Equation 1 that a partial converse holds in low dimensions. For any feasible $(w, \lambda)$, the set of $w'$ such that **(a)** $(w', \lambda)$ is feasible, and **(b)** $\|w\|_1 = \|w'\|_1$, has the form $[0, 1]^n \cap V$ for some subspace $V \subseteq \mathbb{R}^n$ of codimension at most $d + 1$. Thus, there is some $w' \in [0, 1]^n \cap V$ with at most $d + 1$ non-integral weights. If $\text{Stability}(X, y) = \alpha n$, then $w'$ witnesses that the first coordinate of the OLS regressor can be zeroed out by downweighting at most $\alpha n + d + 1$ samples.

## 4 OVERVIEW OF ALGORITHMS

**An exact algorithm.** Our main tool for Theorem 1.2 is the following special case of an important result due to Renegar (1992) on solving quantified polynomial systems of inequalities:

**Theorem 4.1** (Renegar (1992)). *Given an expression*

$$\forall x \in \mathbb{R}^{n_1} : \exists y \in \mathbb{R}^{n_2} : P(x, y),$$

*where $P(x, y)$ is a system of $m$ polynomial inequalities with maximum degree $d$, the truth value of the expression can be decided in time $(md)^{O(n_1 n_2)}$.*[1]

---

[1]This is in the real number model; a similar statement can be made in the bit complexity model.

Roughly, for a constant number of quantifier alternations, a quantified polynomial system can be decided in time exponential in the number of variables. Unfortunately, a naive formulation of the expression $\mathrm{Stability}(X, y) \leq k$, by direct evaluation of Equation 1, has $n + d - 1$ variables:

$$\exists \lambda \in \mathbb{R}^{d-1}, w \in [0, 1]^n : \sum_{i=1}^{n} w_i \geq n - k \text{ and } X^T(w \star (\tilde{X}\lambda - y)) = 0.$$

Intuitively, it may not be necessary to search over all $w \in [0, 1]^n$; for fixed $\lambda$, the maximum-weight $w$ is described by a simple linear program. Formally, the linear program can be rewritten (Lemma B.1) by the separating hyperplane theorem, so that the overall expression becomes:

$$\exists \lambda \in \mathbb{R}^{d-1} : \forall u \in \mathbb{R}^d : \exists w \in [0, 1]^n : \|w\|_1 \geq n - k \text{ and } \sum_{i=1}^{n} w_i(\langle \tilde{X}_i, \lambda \rangle - y_i)\langle X_i, u \rangle \geq 0. \quad (2)$$

Now, for fixed $\lambda$ and $u$, the maximum-weight $w$ has very simple description: it only depends on the *relative ordering* of the $n$ summands $(\langle \tilde{X}, \lambda \rangle - y_i)\langle X_i, u \rangle$. By classical results on connected components of varieties, since the summands have only $2d - 1$ variables, the number of achievable orderings is only $n^{\Omega(d)}$ rather than $n!$, and the orderings can be enumerated efficiently (Milnor, 1964; Renegar, 1992). This allows the quantifier over $w \in [0, 1]^n$ to be replaced by a quantifier over the $n^{\Omega(d)}$ achievable orderings, after which Theorem 4.1 implies that the overall expression can be decided in time $n^{\Omega(d^3)}$. See Appendix B for details.

**Approximation via partitioning.** Next, we show how to avoid the heavy algorithmic machinery used in the previous result. For Theorem 1.4, the strategy of PARTITIONANDAPPROX is to partition the OLS solution space $\mathbb{R}^{d-1}$ into roughly $n^d$ regions, such that if we restrict $\lambda$ to any one region, the bilinear program which defines the stability can be approximated by a linear program.

Concretely, we start by writing the formulation (1) as

$$n - \mathrm{Stability}(x, y) = \sup_{w \in [0,1]^n, \lambda \in \mathbb{R}^{d-1}} \left\{ \sum_{i \in [n]} w_i \middle| X^T(w \star (\tilde{X}\lambda - y)) = 0 \right\}. \quad (3)$$

This has a nonlinear (and nonconvex) constraint due to the pointwise product between $w$ and the residual vector $\tilde{X}\lambda - y$. Thus, we can introduce the change of variables $g_i = w_i(\langle \tilde{X}_i, \lambda \rangle - y_i)$ for $i \in [n]$. This causes two issues. First, the constraint $0 \leq w_i \leq 1$ becomes $0 \leq g_i/(\langle X_i, \lambda \rangle - y_i) \leq 1$, which is no longer linear. To fix this, suppose that instead of maximizing over all $\lambda \in \mathbb{R}^{d-1}$, we maximize over a region $R \subseteq \mathbb{R}^{d-1}$ where each residual $\langle \tilde{X}, \lambda \rangle - y_i$ has constant sign $\sigma_i$. The constraint $0 \leq w_i \leq 1$ then becomes one of two linear constraints, depending on $\sigma_i$. Let $V_R$ denote the value of Program 3 restricted to $\lambda \in R$. Then with the change of variables, we have that

$$V_R = \sup_{g \in \mathbb{R}^n, \lambda \in R} \left\{ \sum_{i \in [n]} \frac{g_i}{\langle X_i, \lambda \rangle - y_i} \middle| \begin{array}{ll} X^T g = 0 & \\ 0 \leq g_i \leq \langle X_i, \lambda \rangle - y_i & \forall i \in [n] : \sigma_i = 1 \\ \langle X_i, \lambda \rangle - y_i \leq g_i \leq 0 & \forall i \in [n] : \sigma_i = -1 \end{array} \right\},$$

with the convention that $0/0 = 1$. Now the constraints are linear. Unfortunately, (and this is the second issue), the objective is no longer linear. The solution is to partition the region $R$ further: if the region were small enough that every residual $\langle X_i, \lambda \rangle - y_i$ had at most $(1 \pm \epsilon)$-multiplicative variation, then the objective could be approximated to within $1 \pm \epsilon$ by a linear objective.

How many regions do we need? Let $M = \|X\beta^{(0)} - y\|_2$ be the unweighted OLS error. If all the residuals were bounded between $\delta M/\sqrt{n}$ and $M$ in magnitude, for all $\lambda \in \mathbb{R}^{d-1}$, then the regions could be demarcated by $O(n \log_{1+\epsilon}(n/\delta))$ hyperplanes, for a total of $O(n \log_{1+\epsilon}(n/\delta))^d$ regions. Of course, for some $\lambda$, some residuals may be very small or very large. But $(\epsilon, \delta)$-anti-concentration implies that for every $\lambda$, at most $\epsilon n$ residuals are very small, and it can be shown that if $\lambda$ is a weighted OLS solution, the total weight on samples with large residuals is low. Thus, for any region, we can exclude from the objective function the samples with residuals that are not well-approximated within the region, and this only affects the objective by $O(\epsilon n)$.

This gives an algorithm with time complexity $(n\epsilon^{-1} \log(1/\delta))^{d+O(1)}$. To achieve the time complexity in Theorem 1.4, where the $\log(n/\delta)$ is additive rather than multiplicative, we use subsampling. Every

residual is still partitioned by sign, but we multiplicatively partition only a random $\tilde{O}(d/\epsilon)$-size subset of the residuals. Intuitively, most residuals will still be well-approximated in any given region. This can roughly be formalized via a VC dimension argument, albeit with some technical complications. See Section D for details and Appendix J for formal pseudocode of the algorithm.

**Net-based approximation.** The algorithm NETAPPROX for Theorem 1.5 is intuitively the simplest. For any fixed $\lambda \in \mathbb{R}^{d-1}$, Program 1 reduces to a linear program with value denoted $S(\lambda)$. Thus, an obvious approach is to construct a net $\mathcal{N} \subseteq \mathbb{R}^{d-1}$ in some appropriate metric, and compute $\min_{\lambda \in \mathcal{N}} S(\lambda)$. This always upper bounds the stability, but to prove that it's an approximate lower bound, we need $S(\lambda)$ to be Lipschitz under the metric.

The right metric turns out to be

$$d(\lambda, \lambda') = \left\| \frac{\tilde{X}\lambda - y}{\left\| \tilde{X}\lambda - y \right\|_2} - \frac{\tilde{X}\lambda' - y}{\left\| \tilde{X}\lambda' - y \right\|_2} \right\|_2 .$$

Under this metric, $\mathbb{R}^{d-1}$ essentially embeds into a $d$-dimensional subspace of the Euclidean sphere $\mathcal{S}^{n-1}$, and therefore has a $\gamma$-net of size $O(1/\gamma)^d$. Why is $S(\lambda)$ Lipschitz under $d$? First, if $\tilde{X}\lambda - y$ equals $\tilde{X}\lambda' - y$ up to rescaling, then it can be seen from Program 3 that $S(\lambda) = S(\lambda')$. More generally, if the residuals are close up to rescaling, we apply the dual formulation of $S(\lambda)$ from expression (2):

$$n - S(\lambda) = \inf_{u \in \mathbb{R}^d} \sup_{w \in [0,1]^n} \|w\|_1 : \sum_{i=1}^{n} w_i(\langle \tilde{X}_i, \lambda \rangle - y_i)\langle X_i, u \rangle \geq 0.$$

For any $u$, the optimal $w$ for $\lambda$ and $u$ can be rounded to some feasible $w'$ for $\lambda'$ and $u$ without decreasing the $\ell_1$ norm too much, under strong anti-concentration. This shows that $S(\lambda)$ and $S(\lambda')$ are close. See Appendix E for details and Appendix J for formal pseudocode of NETAPPROX.

## 5 EXPERIMENTS

In this section, we apply (modifications of) NETAPPROX and PARTITIONANDAPPROX to several datasets in two and three dimensions. Due to space constraints, we defer detailed discussion of the algorithmic modifications to Appendix I.1; we simply note that the modifications are made to improve practical efficiency and usability. Most saliently, the modified algorithms do not rely on Assumptions A and B: the modified NETAPPROX provides an unconditional *upper bound* on stability (referred to henceforth as "net upper bound"), and the modified PARTITIONANDAPPROX provides an unconditional *lower bound* ("LP lower bound"). As a result, we are able to experimentally verify that our algorithms provide tight (and unconditional) bounds on stability for a variety of datasets.

As a baseline upper bound, we implement the greedy heuristic of Kuschnig et al. (2021) which refines Broderick et al. (2020). We are not aware of any prior work on lower bounding stability, so we implement a simplification of our full lower bound algorithm as a baseline. See Appendix I for implementation details, hyperparameter choices, and discussion of error bars.

### 5.1 SYNTHETIC DATA

**Heterogeneous data.** We start with a simple two-dimensional dataset with two disparate sub-populations, where the greedy baseline fails to estimate the stability but our algorithms give tight estimates. For parameters $n, k$, and $\sigma$, we generate $k$ independent samples $(X_i, y_i)$, where $X_i \in \mathbb{R}^2$ has independent coordinates $X_{i1} \sim N(-1, 0.01)$ and $X_{i2} \sim N(0, 1)$, and $y_i = X_{i1}$. Then, we generate $n - k$ independent samples $(X_i, y_i)$ where $X_{i1} = 0$ and $X_{i2} \sim N(0, 1)$, and $y \sim N(0, 1)$. It always suffices to remove the first subpopulation, so the stability is at most $k$. However, the first subpopulation has small individual influences, because the OLS regressor on the whole dataset can nearly interpolate the first subpopulation. Thus, we expect that the greedy algorithm will fail to notice the first subpopulation, and therefore remove far more than $k$ samples.

Indeed, this is what happens. For $n = 1000$ and $k$ varying from 10 to 500, we compare our net upper bound and LP lower bound with the baselines. As seen in Figure 1, our methods are always better

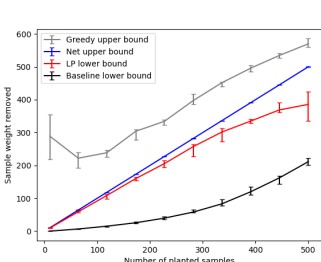

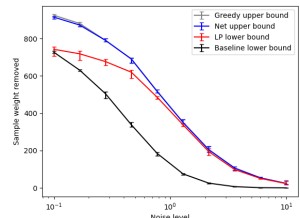

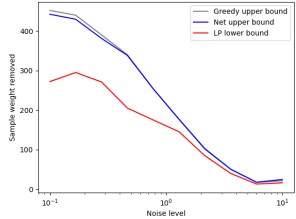

(a) $d = 2$ and $n = 1000$; median (b) $d = 3$ and $n = 500$; one trial of 10 trials for each noise level and for each noise level and algorithm algorithm

Figure 1: Heterogeneous data

Figure 2: Isotropic Gaussian data

than the baselines, and certifiably approximate the stability within a small constant factor. In the regime where $k$ is small, our upper bound outperforms the greedy upper bound by a factor of $30$.

**Covariance shift.** In the previous example, removing $k$ samples caused a pathological change in the sample covariance; it became singular. However, even modest, constant-factor instability in the sample covariance can cause the greedy algorithm to fail; see Appendix I.5 for details.

**Isotropic Gaussian data.** Instability can arise even in homogeneous data, as a result of a low signal-to-noise ratio (Broderick et al., 2020). But when the noise level is low, can we certify stability? For a broad range of noise levels, we experimentally show that this is the case. Specifically, for $d \in \{2, 3\}$ and noise parameter $\sigma$ ranging from 0.1 to 10, we generate $n$ independent samples $(X_i, y_i)_{i=1}^n$ where $X_i \sim N(0, I_d)$ and $y_i = \langle X_i, \mathbb{1} \rangle + N(0, \sigma^2)$. For $d = 2$ and $n = 1000$ (Figure 2a), our LP lower bound is nearly tight with the upper bounds, particularly as the noise level increases (in comparison, the baseline lower bound quickly degenerates towards zero). For $d = 3$ and $n = 500$ (Figure 2b), the bounds are looser for small noise levels but still always within a small constant factor.

## 5.2 BOSTON HOUSING DATASET

The Boston Housing dataset (Harrison Jr & Rubinfeld, 1978; Gilley et al., 1996) consists of data from 506 census tracts of Greater Boston in the 1970 Census. There are 14 real-valued features, one of which—the median house value in USD 1000s—we designate as the response. Unfortunately the entire set of features is too large for our algorithms, so for our experiments we pick various subsets of two or three features to use as covariates.

**A Tale of Two Datasets.** We exemplify our results with two particular feature subsets. First, we investigate the effect of zn (percentage of residential land zoned for large lots) on house values, controlling for rm (average number of rooms per home) and rad (highway accessibility index) but no bias term. On the entire dataset, we find a modest positive effect: the estimated coefficient of zn is roughly $0.06$. Both the greedy heuristic and our net algorithm find subsets of just $8\%$ of the data (38-40 samples) which, if removed, would nullify the effect. But is this tight, or could there be a much smaller subset with the same effect? Our LP lower bound **certifies that removing at least** $22.4$ **samples is necessary**.

Second, we investigate the effect of zn on house values, this time controlling only for crim (per capita crime rate). Our net algorithm finds a subset of just $27\%$ of the data which was driving the effect, and the LP lower bound certifies that the stability is at least $8\%$. But this time, the greedy algorithm removes $90\%$ of the samples, a clear failure. What happened? Plotting zn against crim reveals a striking heterogeneity in the data: $73\%$ of the samples have zn $= 0$, and the remaining $27\%$ of the samples (precisely those removed by the net algorithm) have crim $< 0.83$, i.e. very low crime rates. As in the synthetic example, this heterogeneity explains the greedy algorithm's failure. But heterogeneity is very common in real data: in this case, it's between the city proper and the suburbs, and in fact the OLS regressors of these two subpopulations on all 13 features are markedly different

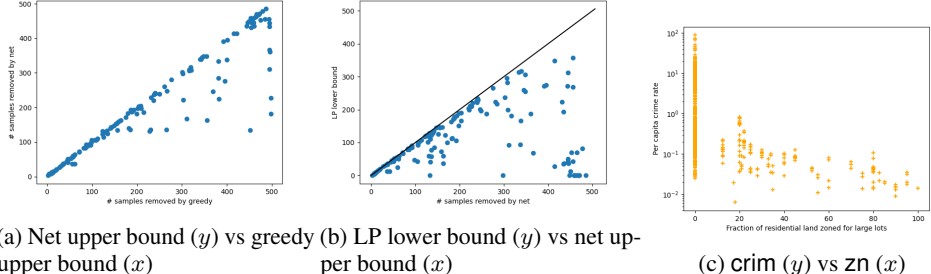

(a) Net upper bound ($y$) vs greedy upper bound ($x$)

(b) LP lower bound ($y$) vs net upper bound ($x$)

(c) crim ($y$) vs zn ($x$)

Figure 3: Results from Boston Housing dataset. Figure (a) plots the net upper bounds on the $y$-axis against the greedy upper bounds on the $x$-axis; Figure (b) plots the LP lower bounds on the $y$-axis against the net upper bounds on the $x$-axis. In both (a) and (b), each mark corresponds to one of the 156 feature pairs. Figure (c) plots the feature zn against the feature crim (on log scale); each mark is one of the 506 datapoints.

(Appendix I.6). Thus, it's important to have algorithms with provable guarantees for detecting when heterogeneity causes (or doesn't cause) unstable conclusions.

**All-feature-pairs analysis.** To be thorough, we also apply our algorithms to all 156 ordered pairs of features. For each pair, we regress the response (i.e. median house value) against the two features by Ordinary Least Squares, and we use our algorithms on this 2-dimensional dataset to estimate how many samples need to be removed to nullify the effect of the first feature on the response. We also compare to the greedy upper bound. See Figure 3 for a perspective on the results. In each figure, each point corresponds to the results of one dataset. The left figure plots the net upper bound against the greedy upper bound: we can see that our net algorithm substantially outperforms the greedy heuristic on some datasets (i.e. finds a much smaller upper bound) and never performs much worse. The right figure plots the LP lower bound against the net upper bound (along with the line $y = x$). For a majority of the datasets, the upper bound and lower bound are close. Concretely, for 116 of the 156 datasets, **we certifiably estimate the stability up to a factor of two** – some are sensitive to removing less than 10 samples, and some are stable to removing even a majority of the samples.

## 6    CONCLUSIONS

In this work, we studied efficient estimation of the stability of OLS regressions to removing subsets of the training data. We showed that in low dimensions the problem is both theoretically and experimentally tractable, whereas in high dimensions exact computation of the stability likely requires exponential time. However, this is only the beginning of the story. Most immediately, since our lower bound algorithm takes time $n^{\Omega(d)}$, our experiments were limited to no more than three dimensions. Certifying stability of OLS regressions from e.g. recent econometric studies may require additional heuristics or insights (e.g. developing a fixed-parameter tractable lower bound algorithm). Beyond that, identifying reasonable assumptions under which exponential dependence on dimension can be entirely circumvented is another valuable direction for future work.

Of course, machine learning extends far beyond linear regression, and for more and more complex and opaque models, stability certification is all the more crucial as a tool for enhancing trustworthiness. Certainly, OLS is important in its own right, but inasmuch as it is a key building block in more complex machine learning systems (from regression trees (Loh, 2011) to generative adversarial networks (Mao et al., 2017) and policy iteration in linear MDPs (Lagoudakis & Parr, 2003)), our work on estimating stability of OLS is also a first step towards estimating stability for these systems.

Finally, we remark that care must be taken when interpreting stability in practice. Large stability may increase trust in a model's parameters or predictions, but it does not mean that conclusions drawn from the model are "correct." Conversely, even if the stability is small, the conclusions may still be useful, with the caveat that they may be driven by a small sub-population. Understanding whether this heterogeneity is problematic or not is context-dependent, and is a separate but important issue.

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

## A    FURTHER RELATED WORK

**Local and global sensitivity metrics.**    Post-hoc evaluation of the sensitivity of a statistical inference to various types of model misspecification has long been recognized as an important research direction. Within this area, there is a distinction between *local* sensitivity metrics, which measure the sensitivity of the inference to infinitesimal misspecifications of the assumed model $M_0$ (e.g. Polasek (1984); Castillo et al. (2004); Belsley et al. (1980)), and *global* sensitivity metrics, which measure the set of possible inferences as the model ranges in some fixed set $\mathcal{M}$ around $M_0$ (e.g. Leamer (1984); Tanaka et al. (1989); Černỳ et al. (2013)). For OLS in particular, there is a well-established literature on the influences of individual data points (Cook, 1977; Chatterjee & Hadi, 1986), which falls under local sensitivity analysis, since deleting a single data point is an infinitesimal perturbation to a dataset of size $n$ as $n \to \infty$. In contrast, identifying jointly influential subsets of the data (the "global" analogue) has been a long-standing challenge due to computational issues (see e.g. page 274 of Belsley et al. (1980)). Existing approaches typically focus on identifying outliers in a generic sense rather than with respect to a specific inference (Hadi & Simonoff, 1993), or study computationally tractable variations of deletion (e.g. constant-factor reweighting (Leamer, 1984)).

**Robustified estimators.**    Ever since the work of Tukey and Huber, one of the central areas of statistics has been robustifying statistical estimators to be resilient to outliers (see, e.g. Huber (2004)). While a valuable branch of research, we view robust statistics as incomparable if not orthogonal to post-hoc sensitivity evaluation, for three reasons. First, samples that *drive* the conclusion (in the sense that deleting them would nullify the conclusion) are not synonymous with outliers: removing an outlier that works against the conclusion only makes the conclusion stronger. Indeed, outlier-trimmed datasets are not necessarily finite-sample robust (Broderick et al., 2020). Rather, finite-sample stability (along with the $s$-value (Gupta & Rothenhäusler, 2021)), in the regime where a constant fraction of samples is removed, may be thought of as a measure of resilience to heterogeneity and distribution shift.

Second, it is unreasonable to argue that using robustified estimators obviates the need for sensitivity evaluation. Robust statistics has seen a recent algorithmic revival, with the development of computationally efficient estimators, for problems such as linear regression, that are robust in the strong contamination model (e.g. Klivans et al. (2018); Diakonikolas et al. (2019); Bakshi & Prasad (2021)). However, even positing that the strong contamination model is correct, estimation guarantees for these algorithms require strong, unverifiable (and unavoidable (Klivans et al., 2018)) assumptions about the uncorrupted data, such as hypercontractivity. Sensitivity analyses should support modeling assumptions, not depend upon them.

Third and perhaps most salient, classical estimators such as OLS are ubiquitous in practice, despite the existence of robust estimators. This alone justifies sensitivity analysis of the resulting scientific conclusions.

**Distributionally robust optimization.**    A recent line of work in machine learning (Sinha et al., 2017; Duchi & Namkoong, 2018; Cauchois et al., 2020; Jeong & Namkoong, 2020) suggests that the lack of resilience of Empirical Risk Minimization to distribution shift can be mitigated by minimizing the supremum of risks with respect to distributions near the empirical training distribution (under e.g. Wasserstein distance or an $f$-divergence). Again, this approach of robustifying the estimator is valuable but incomparable to sensitivity analysis.

## B    PROOF OF THEOREM 1.2

In this section, we show how to exactly compute the stability of a $d$-dimensional dataset in time $n^{O(d^3)}$, proving Theorem 1.2. Our main tool is Theorem 4.1, a special case of an important result due to Renegar (1992) on solving quantified polynomial systems of inequalities.

The expression $\text{Stability}(X, y) \leq k$ can indeed be written as a polynomial system of (degree-2) equations, with only an $\exists$ quantifier. Unfortunately, the number of variables in this naive formulation is $n + d - 1$ ($n$ for the weights and $d - 1$ for the regressor), which yields an algorithm exponential in $n$. Thus, to take advantage of the above theorem, we need to reformulate the expression with fewer

variables. The following lemma rewrites the stability, via the separation theorem for convex sets, in a form where the variable reduction will become apparent.

**Lemma B.1.** *For any* $(X_i, y_i)_{i=1}^n$ *and* $k \geq 0$*, it holds that* $\mathrm{Stability}(X, y) \leq k$ *if and only if*

$$\exists \lambda \in \mathbb{R}^{d-1} : \forall u \in \mathbb{R}^d : \exists w \in [0,1]^n : \|w\|_1 \geq n - k \wedge \sum_{i=1}^n w_i(\langle \tilde{X}_i, \lambda \rangle - y_i)\langle X_i, u \rangle \geq 0, \quad (4)$$

*where* $\tilde{X} : n \times (d-1)$ *is the matrix with columns* $(X^T)_2, \ldots, (X^T)_d$.

*Proof.* From formulation (1) of the stability, we know that $\mathrm{Stability}(X, y) \leq k$ if and only if

$$\exists \lambda \in \mathbb{R}^{d-1} : \exists w \in [0,1]^n : \|w\|_1 \geq n - k \wedge X^T(w \star (\tilde{X}\lambda - y)) = 0.$$

Fix $\lambda \in \mathbb{R}^{d-1}$. Define the set

$$D(n-k) = \left\{ (w \star (\tilde{X}\lambda - y)) : w \in [0,1]^n \wedge \|w\|_1 \geq n - k \right\}.$$

We are interested in the predicate $D(n-k) \cap \ker(X^T) \neq \emptyset$, or equivalently $0 \in D(n-k) + \ker(X^T)$. Observe that $D(n-k)$ is convex, since $w$ ranges over a convex set. Thus, by the separation theorem for a point and a convex set, $0 \in D(n-k) + \ker(X^T)$ if and only if for every $v \in \mathbb{R}^n$, we have $\sup_{x \in D(n-k) + \ker(X^T)} \langle v, x \rangle \geq 0$. If $v$ is not orthogonal to $\ker(X^T)$, then the inner product can be made arbitrarily large. Thus, it suffices to restrict to $v \in \mathrm{span}(X^T)$, in which case the supremum is simply over $x \in D(n-k)$. That is, $0 \in D(n-k) \cap \ker(X^T)$ if and only if

$$\forall u \in \mathbb{R}^d : \exists w \in [0,1]^n : \|w\|_1 \geq n - k \wedge \left\langle Xu, (w \star (\tilde{X}\lambda - y)) \right\rangle \geq 0.$$

Quantifying over $\lambda$, we get the claimed expression. $\qquad\square$

The expression in Lemma B.1 still has $O(n)$ variables. However, we can now actually eliminate the variable $w$ at the cost of increasing the number of equations. This is because the optimal $w$ for fixed $\lambda$ and $u$ only depends on the relative order of the terms $(\langle \tilde{X}_i, \lambda \rangle - y_i)\langle X_i, u \rangle$. We make the following definition:

**Definition B.2.** For any $\lambda \in \mathbb{R}^{d-1}$ and $u \in \mathbb{R}^d$, let $\pi(\lambda, u)$ be the unique permutation on $[n]$ such that for all $1 \leq i \leq n - 1$,

$$(\langle \tilde{X}_{\pi_i}, \lambda \rangle - y_{\pi_i})\langle X_{\pi_i}, u \rangle \geq (\langle \tilde{X}_{\pi_{i+1}}, \lambda \rangle - y_{\pi_{i+1}})\langle X_{\pi_{i+1}}, u \rangle,$$

and such that equality implies $\pi_i < \pi_{i+1}$. Let $\Pi = \{\pi(\lambda, u) : \lambda \in \mathbb{R}^{d-1}, u \in \mathbb{R}^d\}$.

Then it can be seen that for fixed $\lambda$ and $u$, the optimal choice of $w$ has coefficients 1 on $\pi(\lambda, u)_1, \ldots, \pi(\lambda, u)_{\lfloor n-k \rfloor}$, and coefficient $n - k - \lfloor n - k \rfloor$ for $\pi(\lambda, u)_{\lfloor n-k \rfloor + 1}$: if there is any feasible $w$ which makes the sum non-negative, then this choice of $w$ makes the sum non-negative as well. Denoting this vector by $w(\pi(\lambda, u))$, we have that in Equation 4 it suffices to restrict to $w \in \{w(\pi) : \pi \in \Pi\}$.

A priori, the number of achievable permutations could be $n!$, in which case we would not have gained anything. However, because $\pi(\lambda, u)$ is defined by low-degree polynomials in only $2d - 1$ variables, we can actually show that $|\Pi|$ is at most exponential in $d$, using the following result:

**Theorem B.3** (Sign Partitions (Milnor, 1964; Renegar, 1992)). *Let* $g_1, \ldots, g_m : \mathbb{R}^n \to \mathbb{R}$ *be arbitrary polynomials each with total degree at most* $d$. *Let* $\mathsf{SG}(g)$ *be the set of vectors* $\sigma \in \{-1, 0, 1\}^m$ *such that* $\sigma$ *is an achievable sign vector, i.e. there exists some* $x \in \mathbb{R}^n$ *with* $\mathrm{sign}(g_i) = \sigma_i$ *for all* $i \in [m]$. *Then* $|\mathsf{SG}(g)| \leq (md)^{O(n)}$. *Moreover,* $\mathsf{SG}(g)$ *can be enumerated in time* $(md)^{O(n)}$.

Putting everything together, we have the following theorem, which proves Theorem 1.2.

**Theorem B.4.** *For any permutation* $\pi$ *on* $[n]$*, define* $w(\pi) \in [0,1]^n$ *by*

$$w(\pi)_{\pi_i} = \begin{cases} 1 & \text{if } i \leq n - k \\ n - k - \lfloor n - k \rfloor & \text{if } i = n - k + 1 \\ 0 & \text{otherwise} \end{cases}.$$

*Then for any $k \in [0, n]$, it holds that $\mathrm{Stability}(X, y) > k$ if and only if*

$$\forall \lambda \in \mathbb{R}^{d-1} : \exists u \in \mathbb{R}^d : \forall \pi \in \Pi : \sum_{i=1}^n w(\pi)_i (\langle \tilde{X}_i, \lambda \rangle - y_i) \langle X_i, u \rangle < 0. \tag{5}$$

*Moreover, $\Pi$ can be enumerated in time $n^{O(d)}$. Thus, the expression $\mathrm{Stability}(X, y) > k$ can be decided in time $n^{O(d^3)}$.*

*Proof.* Fix $\lambda \in \mathbb{R}^{d-1}$ and $u \in \mathbb{R}^d$. If

$$\exists \pi \in \Pi : \sum_{i=1}^n w(\pi)_i (\langle \tilde{X}_i, \lambda \rangle - y_i) \langle X_i, u \rangle \geq 0, \tag{6}$$

then because $\|w(\pi)\|_1 \geq n - k$, we obviously get

$$\exists w \in [0, 1]^n : \|w\|_1 \geq n - k \wedge \sum_{i=1}^n w_i (\langle \tilde{X}_i, \lambda \rangle - y_i) \langle X_i, u \rangle \geq 0. \tag{7}$$

Conversely, if (6) is false, then in particular $w(\pi(\lambda, u))$ produces a negative sum $\sum_{i=1}^n w(\pi(\lambda, u))_i (\langle \tilde{X}_i, \lambda \rangle - y_i) \langle X_i, u \rangle$. But by construction, $w(\pi(\lambda, u))$ maximizes this sum, over all $w \in [0, 1]^n$ with $\|w\|_1 = n - k$. Therefore no weight vector with $\ell_1$ norm exactly $n - k$ produces a nonnegative sum, and increasing the norm cannot help. Thus, (7) and (6) are equivalent. Quantifying over $\lambda$ and $u$, we have $\mathrm{Stability}(X, y) \leq k$ if and only if

$$\exists \lambda \in \mathbb{R}^{d-1} : \forall u \in \mathbb{R}^d : \exists \pi \in \Pi : \sum_{i=1}^n w(\pi)_i (\langle \tilde{X}_i, \lambda \rangle - y_i) \langle X_i, u \rangle \geq 0.$$

Taking the negation yields expression (5). If we can compute $\Pi$, then this expression is a $\forall \exists$-system of polynomial inequalities with $2d - 1$ variables and $|\Pi|$ degree-2 inequalities, so by Theorem 4.1 it can be decided in time $|\Pi|^{O(d^2)}$. It remains to show that $\Pi$ can be enumerated in time $n^{O(d)}$ (which bounds $|\Pi|$).

For any $i, j \in [n]$ with $i < j$ define the polynomial

$$f_{i,j}(\lambda, u) = (\langle \tilde{X}_i, \lambda \rangle - y_i) \langle X_i, u \rangle - (\langle \tilde{X}_j, \lambda \rangle - y_j) \langle X_j, u \rangle.$$

For any $\lambda$ and $u$, the permutation $\pi(\lambda, u)$ is determined by the signs of the polynomials $\{f_{i,j}\}_{i<j}$ at $(\lambda, u)$. But by Theorem B.3, the set of sign vectors can be computed in time $n^{O(d)}$. So $\Pi$ can be found in time $n^{O(d)}$ as well. $\square$

## C  PROOF OF THEOREM 1.3

In this section, we prove Theorem 1.3. That is, we show that exactly computing the stability requires $n^{\Omega(d)}$ time under the Exponential Time Hypothesis, by a simple reduction from the *Maximum Feasible Subsystem* problem. This latter problem is already known to take $n^{\Omega(d)}$ time in $d$ dimensions under the Exponential Time Hypothesis:

**Theorem C.1** (Theorem 13 in Giannopoulos et al. (2009))**.** *Suppose that there is an $n^{o(d)}$-time algorithm for the following problem: given $n$ vectors $v_1, \ldots, v_n \in \mathbb{R}^d$, real numbers $c_1, \ldots, c_n \in \mathbb{R}$, and an integer $0 \leq k \leq n$, determine whether*

$$\max_{\lambda \in \mathbb{R}^d} \sum_{i=1}^n \mathbb{1}[\langle v_i, \lambda \rangle = c_i] \geq k.$$

*Then the Exponential Time Hypothesis is false.*

From this, it can be easily seen that (exactly) computing stability also requires $n^{\Omega(d)}$ time.

**Theorem C.2.** *Suppose that there is an $n^{o(d)}$-time algorithm for the following problem: given $X_1, \ldots, X_n \in \mathbb{R}^d$ and $y_1, \ldots, y_n \in \mathbb{R}$, as well as an integer $0 \leq k \leq n$, determine whether*

$$\mathrm{Stability}(X, y) \leq n - k.$$

*Then the Exponential Time Hypothesis is false.*

*Proof.* We reduce to Maximum Feasible Subsystem. Given $v_1, \ldots, v_n \in \mathbb{R}^d$ and $c_1, \ldots, c_n \in \mathbb{R}$, define $X_i = (c_i, v_i) \in \mathbb{R}^{d+1}$ and $y_i = c_i$. Then the regressor $e_1 = (1, 0, \ldots, 0) \in \mathbb{R}^{d+1}$ perfectly fits the data set, i.e.

$$\sum_{i=1}^{n} (\langle X_i, e_1 \rangle - y_i)^2 = 0.$$

Thus, for any $w \in [0, 1]^n$,

$$\mathsf{OLS}(X, y, w) = \left\{ \beta \in \mathbb{R}^{d+1} : \sum_{i=1}^{n} w_i(\langle X_i, \beta \rangle - y_i)^2 = 0 \right\}.$$

Suppose that $\mathrm{Stability}(X, y) \leq n - k$. Then there is some $w \in [0, 1]^n$ and $\beta_1 = 0$ with $\|w\|_1 \geq k$ and $\sum_{i=1}^{n} w_i(\langle X_i, \beta \rangle - y_i)^2 = 0$. Let $S \subseteq [n]$ be the support of $w$; then $|S| \geq k$. Moreover for every $i \in S$, it holds that $\langle X_i, \beta \rangle - y_i = 0$. Since $\beta_1 = 0$, by definition of $X_i$ and $y_i$, this implies that $\langle v_i, \beta_{2:d+1} \rangle = c_i$. Thus,

$$\sum_{i=1}^{n} \mathbb{1}[\langle v_i, \beta_{2:d+1} \rangle = c_i] \geq k.$$

Conversely, suppose that there exists some $\lambda \in \mathbb{R}^d$ and set $S \subseteq [n]$ of size $k$ such that $\langle v_i, \lambda \rangle = c_i$ for all $i \in S$. Define $w = \mathbb{1}_S \in [0, 1]^n$, and define $\beta = (0, \lambda)$. Then it is clear that

$$\sum_{i=1}^{n} w_i(\langle X_i, \beta \rangle - y_i)^2 = \sum_{i \in S} (\langle v_i, \lambda \rangle - c_i)^2 = 0.$$

Thus, $\beta \in \mathsf{OLS}(X, y, w)$, so $\mathrm{Stability}(X, y) \leq n - k$. This completes the reduction. $\square$

**Remark C.3.** Maximum Feasible Subsystem does have an $\epsilon n$-additive approximation algorithm in time $\tilde{O}((d/\epsilon)^d)$, by subsampling. Thus, proving $n^{\Omega(d)}$-hardness for $\epsilon n$-approximation of stability would require a different technique.

# D  PROOF OF THEOREM 1.4

In this section, we prove Theorem 1.4. The main idea of PARTITIONANDAPPROX is that under Assumption A, we can approximate the stability by partitioning $\mathbb{R}^{d-1}$ into roughly $n^d$ regions and solving a linear program on each region. See Appendix J for the complete algorithm.

## D.1  PARTITIONING SCHEME

Given samples $(X_i, y_i)_{i=1}^n$, let $M = \left\| X\beta^{(0)} - y \right\|_2$, where $\beta^{(0)} \in \mathsf{OLS}(X, y, \mathbb{1})$. Let $S \subseteq [n]$ be a uniformly random size-$m$ subset of $[n]$. Let $R_1, \ldots, R_p$ be the closed connected subsets of $\mathbb{R}^{d-1}$ cut out by the following set of equations $\mathcal{E}$:

$$\begin{cases} \langle \tilde{X}_i, \lambda \rangle - y_i = 0 & \forall i \in [n] \\ \langle \tilde{X}_i, \lambda \rangle - y_i = \sigma M & \forall i \in [n], \forall \sigma \in \{-1, 1\} \\ \langle \tilde{X}_i, \lambda \rangle - y_i = \sigma \delta M / \sqrt{n} & \forall i \in [n], \forall \sigma \in \{-1, 1\} \\ \langle \tilde{X}_i, \lambda \rangle - y_i = \sigma(1 + \epsilon)^k \delta M / \sqrt{n} & \forall i \in S, \forall \sigma \in \{-1, 1\}, \forall 0 \leq k \leq \lceil \log_{1+\epsilon}(\sqrt{n}/\delta) \rceil \end{cases}$$

Formally, we define a region for every feasible assignment of equations to $\{=, <, >\}$, and then replace each strict inequality by a non-strict inequality, so that the region is closed. First, we observe that the number of regions is not too large, and in fact we can enumerate the regions efficiently.

**Lemma D.1.** *Each region $R_i$ is the intersection of $O(|\mathcal{E}|)$ linear equalities or inequalities. Moreover, the regions $R_1, \ldots, R_p$ can be enumerated in time $O(|\mathcal{E}|)^{d+O(1)}$.*

*Proof.* Order the set of equations $\mathcal{E}$ arbitrarily. We recursively construct the set of regions demarcated by the first $t$ equations. For each such region, we solve a linear program to check if the $t + 1$th hyperplane intersects the interior of the region. If so, we split the region according to the sign of the $t + 1$th hyperplane. The overall time complexity of this procedure is $O(p \cdot \text{poly}(n, |\mathcal{E}|))$, where $p$ is the final number of regions. But the number of regions which can be cut out by $t$ hyperplanes in $\mathbb{R}^d$ is at most $O(t)^d$ by standard arguments. $\qquad\square$

Next, we argue that even though we only multiplicatively partitioned a small subset of the residuals, with high probability most residuals are well-approximated. More precisely, we show that for the (random) region $R$ containing a fixed point $\lambda^*$, with high probability, for every other $\lambda$ in the region, most residuals at $\lambda$ are multiplicatively close to the corresponding residuals at $\lambda^*$. This can be proven by the generalization bound for function classes with low VC dimension:

**Theorem D.2** (Theorem 3.4 in Kearns & Vazirani (1994)). *Let $\mathcal{X}$ be a set. Let $\mathcal{C} \subset \{0, 1\}^{\mathcal{X}}$ be a binary concept class with VC dimension $d$. Let $\mathcal{D}$ be a distribution on $\mathcal{X} \times \{0, 1\}$. Let $\epsilon, \delta > 0$ and pick $m \in \mathbb{N}$ satisfying*

$$m \geq C_1 \left( \frac{d}{\epsilon} \log \frac{1}{\epsilon} + \frac{1}{\epsilon} \log \frac{1}{\delta} \right)$$

*for a sufficiently large constant $C_1$. Let $(x_i, y_i)_{i \in [m]}$ be $m$ independent samples from $\mathcal{D}$. Then with probability at least $1 - \delta$, the following holds. For all $h \in \mathcal{C}$ with $h(x_i) = y_i$ for all $i \in [m]$,*

$$\Pr_{(x,y) \sim \mathcal{D}}[h(x) \neq y] \leq \epsilon.$$

Specifically, for any $\lambda \in \mathbb{R}^{d-1}$, we define a function $f_\lambda : [n] \to \{0, 1\}$ as the indicator function of samples for which the residual at $\lambda$ is close to the residual at $\lambda^*$. Then the region containing $\lambda^*$ is precisely the set of functions which perfectly fit $S \times \{1\}$, and the generalization bound implies that all such functions fit most of $[n] \times 1$, which is what we wanted to show. The following lemma formalizes this argument, with some additional steps to deal with very small and very large residuals.

**Lemma D.3.** *Let $\lambda^* \in \mathbb{R}^{d-1}$ and let $\eta > 0$. Let $R$ be the region containing $\lambda^*$. Let $B_M \subseteq [n]$ be the set of $i \in [n]$ such that $|\langle \tilde{X}_i, \lambda^* \rangle - y_i| > M$, and let $B_{\delta M} \subseteq [n]$ be the set of $i \in [n]$ such that $|\langle \tilde{X}_i, \lambda^* \rangle - y_i| \leq \delta M / \sqrt{n}$. If*

$$m \geq \frac{C}{\epsilon} \left( d \log \frac{1}{\epsilon} + \log \frac{1}{\eta} \right)$$

*for an absolute constant $C$, then with probability at least $1 - \eta$ over the choice of $S$, the following holds. For every $\lambda \in R$, the number of $i \in [n] \setminus (B_M \cup B_{\delta M})$ such that*

$$\left| \frac{|\langle \tilde{X}_i, \lambda \rangle - y_i|}{|\langle \tilde{X}_i, \lambda \rangle - y_i|} - 1 \right| > \epsilon$$

*is at most $\epsilon n$.*

*Proof.* Let $\sigma \in \{-1, 0, 1\}^n$ be defined by $\sigma_i = \text{sign}(\langle \tilde{X}_i, \lambda^* \rangle - y_i)$. Note that by construction of the regions, $\text{sign}(\langle \tilde{X}_i, \lambda \rangle - y_i) = \sigma_i$ for every $\lambda \in R$.

Let $S' = S \cap ([n] \setminus (B_M \cup B_{\delta M}))$. If $n' := |[n] \setminus (B_M \cup B_{\delta M})| \leq \epsilon n$, then the lemma statement is trivially true. Otherwise, by a Chernoff bound,

$$\Pr \left[ |S'| \geq \frac{n'}{2n} m \right] \geq 1 - \exp(-\Omega(n'm/n)) \geq 1 - \eta/3.$$

Condition on the event $|S'| \geq n'm/(2n)$. Then $S'$ is uniform over size-$m'$ subsets of $[n] \setminus (B_M \cup B_{\delta M})$. Define $\mathcal{X} = [n]$, and define a concept class $\mathcal{C} = \{h_\lambda : \mathcal{X} \to \{0, 1\} | \lambda \in \mathbb{R}^{d-1}\}$ as the set of binary functions

$$h_\lambda(i) = \mathbb{1} \left[ \sigma_i \left( \left\langle \tilde{X}_i, \lambda - (1 + \epsilon)\lambda^* \right\rangle + \epsilon y_i \right) \leq 0 \right].$$

Let $\mathcal{D}$ be the distribution on $\mathcal{X} \times \{0,1\}$ where $(i,s) \sim \mathcal{D}$ has $i \sim \text{Unif}([n] \setminus (B_M \cup B_{\delta M}))$ and $s = 1$. Then $S' \times \{1\}$ consists of at least $n'm/(2n)$ independent samples from $\mathcal{D}$. For any $\lambda \in R$, we claim that the function $h_\lambda$ fits $S' \times \{1\}$ perfectly. Indeed, for any $i \in S'$, we know that $\delta M/\sqrt{n} \leq \sigma_i(\langle \tilde{X}_i, \lambda^* \rangle - y_i) \leq M$, since $i \notin B_M \cup B_{\delta M}$. So by construction of the regions, it holds that

$$\frac{1}{1+\epsilon} \leq \frac{\sigma_i(\langle \tilde{X}_i \lambda \rangle - y_i)}{\sigma_i(\langle \tilde{X}_i \lambda^* \rangle - y_i)} \leq 1 + \epsilon.$$

From the right-hand side of the equation, we precisely get that $h_\lambda(i) = 1$, as claimed. By Lemma D.4, we have $\text{vc}(\mathcal{C}) \leq d$. Thus, since

$$|S'| \geq \frac{n'm}{2n} \geq \frac{Cd}{(2n/n')\epsilon} \log \frac{1}{\epsilon} + \frac{C}{(2n/n')\epsilon} \log \frac{1}{\eta},$$

if $C$ is a sufficiently large absolute constant, we can apply Theorem D.2 with failure parameter $\epsilon' := n\epsilon/(2n')$ to get that with probability at least $1 - \eta/3$,

$$\sup_{\lambda \in R} \Pr_{(i,s) \sim \mathcal{D}}[h_\lambda(i) \neq s] \leq \epsilon'.$$

Equivalently, for every $\lambda \in R$, the number of $i \in [n] \setminus (B_M \cup B_{\delta M})$ such that $\sigma_i(\langle \tilde{X}_i, \lambda \rangle - y_i) > (1+\epsilon)\sigma_i(\langle \tilde{X}_i, \lambda \rangle - y_i)$ is at most $\epsilon'n' = \epsilon n/2$. An identical argument with the binary functions

$$g_\lambda(i) = \mathbb{1}\left[\sigma_i\left(\left\langle \tilde{X}_i, \lambda - (1-\epsilon)\lambda^* \right\rangle - \epsilon y_i\right) \geq 0\right]$$

proves that with probability at least $1 - \eta/3$, for every $\lambda \in R$, the number of $i \in [n] \setminus (B_M \cup B_{\delta M})$ such that $\sigma_i(\langle \tilde{X}_i, \lambda \rangle - y_i) < (1-\epsilon)\sigma_i(\langle \tilde{X}_i, \lambda \rangle - y_i)$ is at most $\epsilon n/2$. Overall, by the union bound (taking into account the event that $|S'| < n'm/(2n)$), with probability at least $1 - \eta$ it holds that for every $\lambda \in R$, the number of $i \in [n] \setminus (B_M \cup B_{\delta M})$ such that either $\sigma_i(\langle \tilde{X}_i, \lambda \rangle - y_i) > (1+\epsilon)\sigma_i(\langle \tilde{X}_i, \lambda \rangle - y_i)$ or $\sigma_i(\langle \tilde{X}_i, \lambda \rangle - y_i) < (1-\epsilon)\sigma_i(\langle \tilde{X}_i, \lambda \rangle - y_i)$ is at most $\epsilon n$ as claimed. □

It remains to bound the VC dimension of the function class.

**Lemma D.4.** *For any $n, d > 0$ and any $(X_i, y_i)_{i \in [n]} \subset \mathbb{R}^d \times \mathbb{R}$, the VC dimension of the concept class $\mathcal{C} = \{h_\lambda : [n] \to \{0,1\} | \lambda \in \mathbb{R}^d\}$, where*

$$h_\lambda(i) = \mathbb{1}[\langle X_i, \lambda \rangle + y_i \leq 0],$$

*is at most $d + 1$.*

*Proof.* Note that extending the domain of the concepts cannot decrease the VC dimension. Thus, if we define $\mathcal{C}' = \{h'_\lambda : \mathbb{R}^d \times \mathbb{R} \to \{0,1\} : \lambda \in \mathbb{R}^d\}$ by

$$h'_\lambda(X, y) = \mathbb{1}[\langle X, \lambda \rangle + y \leq 0],$$

then $\text{vc}(\mathcal{C}') \geq \text{vc}(\mathcal{C})$. But all of the concepts in $\mathcal{C}'$ are affine halfspaces in $d + 1$ dimensions, so $\text{vc}(\mathcal{C}') \leq d + 1$. □

### D.2 ALGORITHM

For every region $R$ we can identify some arbitrary representative $\lambda_0(R) \in R$ and a sign pattern $\sigma \in \{-1, 1\}^n$ such that $\sigma_i = 1$ implies $\langle \tilde{X}_i, \lambda \rangle - y_i \geq 0$ for all $\lambda \in R$, and $\sigma_i = -1$ implies $\langle \tilde{X}_i, \lambda \rangle - y_i \leq 0$ for all $\lambda \in R$. Ideally, we want to compute $V = \max_R V_R$ where

$$V_R := \sup_{g \in \mathbb{R}^n, \lambda \in R} \left\{ \sum_{i \in [n]} \frac{g_i}{\langle \tilde{X}_i, \lambda \rangle - y_i} \; \middle| \; \begin{matrix} X^T g = 0 \\ 0 \leq g_i \leq \langle \tilde{X}_i, \lambda \rangle - y_i \quad \forall i \in [n] : \sigma_i = 1 \\ \langle \tilde{X}_i, \lambda \rangle - y_i \leq g_i \leq 0 \quad \forall i \in [n] : \sigma_i = -1 \end{matrix} \right\} \quad (8)$$

However, this is not a linear program. Let $B_M = B_M(R) \subseteq [n]$ be the set of $i \in [n]$ such that $|\langle \tilde{X}_i, \lambda_0(R) \rangle - y_i| > M$, and let $B_{\delta M} = B_{\delta M}(R) \subseteq [n]$ be the set of $i \in [n]$ such that

$|\langle \tilde{X}_i, \lambda_0(R) - y_i| < \delta M/\sqrt{n}$. For each region $R$ we compute

$$(\hat{g}(R), \hat{\lambda}(R)) \tag{9}$$

$$:= \underset{g \in \mathbb{R}^n, \lambda \in R}{\arg\sup} \left\{ \sum_{i \in [n] \setminus (B_M \cup B_{\delta M})} \min\left( \frac{g_i}{\langle \tilde{X}_i, \lambda_0(R) \rangle - y_i}, 1 \right) \middle| \begin{array}{l} X^T g = 0 \\ 0 \leq g_i \leq \langle \tilde{X}_i, \lambda \rangle - y_i \\ \quad \forall i \in [n] : \sigma_i = 1 \\ \langle \tilde{X}_i, \lambda \rangle - y_i \leq g_i \leq 0 \\ \quad \forall i \in [n] : \sigma_i = -1 \end{array} \right\}$$

$$\tag{10}$$

and define

$$\hat{V}(R) = \sum_{i \in [n]} \frac{\hat{g}(R)_i}{\langle \tilde{X}_i, \hat{\lambda}(R) \rangle - y_i}$$

with the convention that $0/0 = 1$. Note that although there is a $\min$ in the objective of Program 10, it is equivalent to a linear program by a standard transformation: we can introduce variables $s_1, \ldots, s_n$ with the constraints $s_i \leq 1$ and $s_i \leq g_i/(\langle X_i, \lambda_0(R) \rangle - y_i)$, and change the objective to maximize $\sum_{i \in [n]} s_i$. That is, the following program also computes $(\hat{g}(R), \hat{\lambda}(R))$:

$$(\hat{g}(R), \hat{\lambda}(R)) \tag{11}$$

$$= \underset{g \in \mathbb{R}^n, \lambda \in R, s \in \mathbb{R}^n}{\arg\sup} \left\{ \sum_{i \in [n] \setminus (B_M \cup B_{\delta M})} s_i \middle| \begin{array}{ll} X^T g = 0 & \\ 0 \leq g_i \leq \langle X_i, \lambda \rangle - y_i & \forall i \in [n] : \sigma_i = 1 \\ \langle X_i, \lambda \rangle - y_i \leq g_i \leq 0 & \forall i \in [n] : \sigma_i = -1 \\ s_i \leq 1 & \forall i \in [n] \setminus (B_M \cup B_{\delta M}) \\ s_i \leq g_i/(\langle X_i, \lambda_0(R) \rangle - y_i) & \forall i \in [n] \setminus (B_M \cup B_{\delta M}) \end{array} \right\}$$

$$\tag{12}$$

**Lemma D.5.** *Let $(w^*, \lambda^*)$ be an optimal solution to Program 1. Suppose that the event of Lemma D.3 holds (with respect to $\lambda^*$). Then either $\max_R B_{\delta M}(R) > \epsilon n$, or*

$$V - 12\epsilon n - 1 \leq \max_R \hat{V}(R) \leq V.$$

*Proof.* For the RHS, observe that for any region $R$, if we define $w \in \mathbb{R}^n$ by $w_i = \hat{g}(R)_i/(\langle X_i, \hat{\lambda}(R) \rangle - y_i)$, with the convention that $0/0 = 1$, then the second and third constraints of Program 10 ensure that $w \in [0, 1]^n$. Additionally, the first constraint ensures that

$$X^T \sum_{i \in [n]} w_i(\langle \tilde{X}_i, \hat{\lambda}(R) \rangle - y_i) = 0.$$

Thus, $(w, \hat{\lambda}(R))$ is feasible for the original problem. This means that $\hat{V}(R) = \|w\|_1 \leq V$.

To prove the lower bound, suppose that $\max_R B_{\delta M}(R) \leq \epsilon n$. Consider the specific region $R$ containing the optimal parameter vector $\lambda^*$ (if there are multiple choose any), and let $B_M = B_M(R)$ and $B_{\delta M} = B_{\delta M}(R)$. Define $g^* \in \mathbb{R}^n$ by $g_i^* = w_i^*(\langle X_i, \lambda^* \rangle - y_i)$. Let $B_{\text{apx}} \subset [n] \setminus (B_M \cup B_{\delta M})$ be the set of $i \in [n] \setminus (B_M \cup B_{\delta M})$ such that

$$\left[ \frac{1}{1+\epsilon}(\langle \tilde{X}_i, \lambda_0(R) \rangle - y_i) > \langle \tilde{X}_i, \lambda^* \rangle - y_i \right] \vee \left[ \langle \tilde{X}_i, \lambda^* \rangle - y_i > (1+\epsilon)(\langle \tilde{X}_i, \lambda_0(R) \rangle - y_i) \right].$$

By Lemma D.3 (applied specifically to $\lambda = \lambda_0(R)$), we know $|B_{\text{apx}}| \leq \epsilon n$. By assumption, we know that $|B_{\delta M}| \leq \epsilon n$. And we know that if $i \in B_M$ then $|\langle \tilde{X}_i, \lambda_0(R) \rangle - y_i| \geq M$, so the same holds for

all $\lambda \in R$ and in particular for $\lambda^*$. Thus

$$\sum_{i \in B_M} w_i^* \leq \frac{1}{M^2} \sum_{i \in B_M} w_i^*(\langle \tilde{X}_i, \lambda^* \rangle - y_i)^2$$

$$\leq \frac{1}{M^2} \sum_{i \in [n]} w_i^*(\langle \tilde{X}_i, \lambda^* \rangle - y_i)^2$$

$$\leq \frac{1}{M^2} \sum_{i \in [n]} w_i^*(\langle X_i, \beta^{(0)} \rangle - y_i)^2$$

$$\leq \frac{1}{M^2} \left\| X\beta^{(0)} - y \right\|_2^2$$

$$= 1$$

where the third inequality is because $\lambda^* \in \mathsf{OLS}(X, y, w^*)$, and the equality is by definition of $M$.
Finally, for any $i \in [n] \setminus (B_M \cup B_{\delta M} \cup B_{\mathrm{apx}})$, we have

$$\min\left( \frac{g_i^*}{\langle X_i, \lambda_0(R) \rangle - y_i}, 1 \right) \geq \min\left( \frac{1}{1+\epsilon} \frac{g_i^*}{\langle X_i, \lambda^* \rangle - y_i}, 1 \right) = \frac{1}{1+\epsilon} \frac{g_i^*}{\langle X_i, \lambda^* \rangle - y_i}.$$

Therefore

$$V = \sum_{i \in [n]} w_i^*$$

$$= \sum_{i \in [n] \setminus (B_M \cup B_{\delta M} \cup B_{\mathrm{apx}})} \frac{g_i^*}{\langle X_i, \lambda^* \rangle - y_i} + \sum_{i \in B_M} \frac{g^*}{\langle X_i, \lambda^* \rangle - y_i} + \sum_{i \in B_{\delta M}} w_i^* + \sum_{i \in B_{\mathrm{apx}}} w_i^*$$

$$\leq (1+\epsilon) \sum_{i \in [n] \setminus (B_M \cup B_{\delta M} \cup B_{\mathrm{apx}})} \min\left( \frac{g_i^*}{\langle X_i, \lambda_0(R) \rangle - y_i}, 1 \right) + 1 + 2\epsilon n$$

$$\leq (1+\epsilon) \sum_{i \in [n] \setminus (B_M \cup B_{\delta M})} \min\left( \frac{g_i^*}{\langle X_i, \lambda_0(R) \rangle - y_i}, 1 \right) + 1 + 2\epsilon n.$$

The sum in the last line above is precisely the objective of Program 10 at $(g^*, \lambda^*)$. Moreover, $(g^*, \lambda^*)$ is feasible for Program 10 because $X^T g^* = X^T(w^* \star (\langle \tilde{X}_i, \lambda^* \rangle - y_i)) = 0$ and $g_i^*/(\langle \tilde{X}_i, \lambda^* \rangle - y_i) = w_i \in [0, 1]$ for all $i \in [n]$. Thus, the optimal solution $(\hat{g}(R), \hat{\lambda}(R))$ satisfies the inequality

$$\sum_{i \in [n] \setminus (B_M \cup B_{\delta M})} \min\left( \frac{g_i^*}{\langle \tilde{X}_i, \lambda_0(R) \rangle - y_i}, 1 \right) \leq \sum_{i \in [n] \setminus (B_M \cup B_{\delta M})} \min\left( \frac{\hat{g}(R)}{\langle \tilde{X}_i, \lambda_0(R) \rangle - y_i}, 1 \right).$$

Finally, let $\hat{B}_{\mathrm{apx}} \subseteq [n]$ be the set of $i \in [n] \setminus (B_M \cup B_{\delta M})$ such that

$$\left[ \frac{1}{(1+\epsilon)^2}(\langle \tilde{X}_i, \lambda_0(R) \rangle - y_i) > \langle \tilde{X}_i, \hat{\lambda}(R) \rangle - y_i \right] \vee \left[ \langle \tilde{X}_i, \hat{\lambda}(R) \rangle - y_i > (1+\epsilon)^2(\langle X_i, \lambda_0(R) \rangle - y_i) \right].$$

By Lemma D.3, the residuals at $\lambda_0(R)$ multiplicatively approximate the residuals at $\lambda^*$ except for $\epsilon n$ samples, and the residuals at $\hat{\lambda}(R)$ also multiplicatively approximate the residuals at $\lambda^*$ except for $\epsilon n$ samples. Thus, $|\hat{B}_{\mathrm{apx}}| \leq 2\epsilon n$, so we have

$$\sum_{i \in [n] \setminus B_M} \min\left( \frac{\hat{g}(R)}{\langle X_i, \lambda_0(R) \rangle - y_i}, 1 \right) = \sum_{i \in [n] \setminus (B_M \cup B_{\delta M} \cup \hat{B}_{\mathrm{apx}})} \min\left( \frac{\hat{g}(R)}{\langle X_i, \lambda_0(R) \rangle - y_i}, 1 \right)$$

$$+ \sum_{i \in B_{\delta M} \cup \hat{B}_{\mathrm{apx}}} \min\left( \frac{\hat{g}(R)}{\langle X_i, \lambda_0(R) \rangle - y_i}, 1 \right)$$

$$\leq \sum_{i \in [n] \setminus (B_M \cup B_{\delta M} \cup \hat{B}_{\mathrm{apx}})} \min\left( \frac{\hat{g}(R)}{\langle X_i, \lambda_0(R) \rangle - y_i}, 1 \right) + 2\epsilon n$$

$$\leq (1+\epsilon)^2 \sum_{i \in [n] \setminus (B_M \cup B_{\delta M} \cup \hat{B}_{\mathrm{apx}})} \frac{\hat{g}(R)}{\langle X_i, \hat{\lambda}(R) \rangle - y_i} + 2\epsilon n$$

$$\leq (1+\epsilon)^2 \hat{V}(R) + 2\epsilon n.$$

We conclude that $V \leq (1+\epsilon)^3 \hat{V}(R) + (1+\epsilon)2\epsilon n + 1 + 2\epsilon n$. Since $\hat{V}(R) \leq n$, simplifying gives $V \leq \hat{V}(R) + 12\epsilon n + 1$ as claimed. $\qquad\square$

Using the above lemma in conjunction with Lemma D.3 immediately gives the desired theorem (from which Theorem 1.4 is a direct corollary).

**Theorem D.6.** *For any $\epsilon, \delta, \eta > 0$, there is an algorithm* PARTITIONANDAPPROX *with time complexity*

$$\left( n + \frac{Cd}{\epsilon^2} \log \frac{n}{\delta} \log \frac{1}{\epsilon\eta} \right)^{d+O(1)}$$

*which, given $\epsilon$, $\delta$, $\eta$, and arbitrary samples $(X_i, y_i)_{i=1}^n$, either outputs $\perp$ or an estimate $\hat{S}$. If the output is $\perp$, then the samples do not satisfy $(\epsilon, \delta)$-anti-concentration (Assumption A). Moreover, the probability that the output is some $\hat{S}$ such that*

$$|\hat{S} - \mathrm{Stability}(X, y)| > 12\epsilon n + 1$$

*is at most $\eta$.*

*Proof.* The algorithm PARTITIONANDAPPROX does the following (see Appendix J for pseudocode). Let $m = C\epsilon^{-1}(d \log \epsilon^{-1} + \log \eta^{-1})$ where $C$ is the constant specified in Lemma D.3, and let $M = \left\| X\beta^{(0)} - y \right\|_2$ where $\beta^{(0)} \in \mathsf{OLS}(X, y, \mathbb{1})$. Let $\mathcal{E}$ be the set of equations described in Section D.1, with respect to a uniformly random subset $S \subseteq [n]$ of size $m$. Let $R_1, \ldots, R_p$ be the closed connected regions cut out by $\mathcal{E}$. By Lemma D.1, we can enumerate $R_1, \ldots, R_p$ in time $O(|\mathcal{E}|)^{d+O(1)}$; each is described by at most $|\mathcal{E}|$ linear constraints. For each $R$, we can find a representative $\lambda_0(R) \in R$ by solving a feasibility LP on $R$, and by solving $n$ LPs on $R$, we can find a sign pattern $\sigma \in \{-1, 1\}^n$ such that $\sigma_i = 1$ implies $\langle \tilde{X}_i, \lambda \rangle - y_i \geq 0$ for all $\lambda \in R$, and $\sigma_i = -1$ implies $\langle \tilde{X}_i, \lambda \rangle - y_i \leq 0$ for all $\lambda \in R$. We also compute $B_M = \{i \in [n] : |\langle \tilde{X}_i, \lambda_0(R)\rangle - y_i| > M\}$ and $B_{\delta M}\{i \in [n] : |\langle \tilde{X}_i, \lambda_0(R)\rangle - y_i| < \delta M/\sqrt{n}\}$. If $|B_{\delta M}| > \epsilon n$, then return $\perp$. Otherwise, compute $(\hat{g}(R), \hat{\lambda}(R))$, the solution to Program 12, and compute $\hat{V}(R) = \sum_{i=1}^n \frac{\hat{g}(R)_i}{\langle \tilde{X}_i, \hat{\lambda}(R)\rangle - y_i}$.

Finally, after iterating through all regions, output $\max_{i \in [p]} \hat{V}(R_i)$.

Correctness follows from Lemmas D.3 and D.5. For each region, the time complexity is $\mathrm{poly}(|\mathcal{E}|)$. As there are $O(|\mathcal{E}|)^d$ regions, the overall time complexity is $O(|\mathcal{E}|)^{d+O(1)}$. Observing that $|\mathcal{E}| = O(n + m\epsilon^{-1} \log(n/\delta))$ completes the proof. $\qquad\square$

# E  PROOF OF THEOREM 1.5

In the previous section, we approximated the nonlinear program (1) by partitioning $\mathbb{R}^{d-1}$ into regions where the program could be approximated by a linear program. This approach had the disadvantage of requiring that in each region, the signs of the residuals $\langle \tilde{X}_i, \lambda \rangle - y_i$ were constant (so that the program could be reparametrized to have linear constraints), which necessitates making $\Omega(n^d)$ regions. In this section, we instead make use of the fact that for fixed $\lambda$, Program 1 is a linear program and therefore efficiently solvable. Our algorithm NETAPPROX is simply to (carefully) choose a finite subset $\mathcal{N} \subset \mathbb{R}^{d-1}$, solve the linear program for each $\lambda \in \mathcal{N}$, and pick the best answer.

The following lemma describes how to compute $\mathcal{N}$, which will be an $\epsilon$-net over $\mathbb{R}^{d-1}$ in an appropriate metric.

**Lemma E.1.** *For any $(X_i, y_i)_{i=1}^n$ and $\gamma > 0$, there is a set $\mathcal{N} \subseteq \mathbb{R}^{d-1}$ of size $|\mathcal{N}| \leq (2\sqrt{d}/\gamma)^d$ such that for any $\lambda \in \mathbb{R}^{d-1}$ with $\tilde{X}\lambda \neq y$, there is some $\lambda' \in \mathcal{N}$ with $\tilde{X}\lambda' \neq y$, and some $\sigma \in \{-1, 1\}$, such that*

$$\left\| \frac{\tilde{X}\lambda - y}{\left\| \tilde{X}\lambda - y \right\|_2} - \frac{\tilde{X}\lambda' - y}{\left\| \tilde{X}\lambda' - y \right\|_2} \right\|_2 \leq \gamma.$$

*Moreover, $\mathcal{N}$ can be computed in time $O(\sqrt{d}/\gamma)^d$.*

*Proof.* Let $A : n \times d$ be the matrix with columns $(X^T)_2, \ldots, (X^T)_d, -y$. Let $d' = \text{rank}(A)$ and let $A = UDV^T$ be the singular value decomposition of $A$, where $D = \text{diag}(s_1, \ldots, s_{d'})$ is the diagonal matrix of nonzero singular values of $A$. Let $\mathcal{M}$ be a "marginally-random" $\gamma$-net over the unit sphere $\mathcal{S}^{d'-1}$ under the $\ell_2$ metric, where by "marginally-random" we mean that $\mathcal{M}$ is chosen from some distribution, and every point of the net has marginal distribution uniform over $\mathcal{S}^{d'-1}$ (e.g. take any fixed $\gamma$-net and apply a uniformly random rotation). Also define $B = VD^{-1}$. Suppose $B_d$ is not identically zero. Then define

$$\mathcal{N} = \left\{ \frac{(Bm)_{1:d-1}}{(Bm)_d} : m \in \mathcal{M} \right\}.$$

Since each $m \in \mathcal{M}$ is a generic unit vector (by the marginally-random property) and $B_d$ is nonzero, we have that all $(Bm)_d$ are nonzero with probability 1, so the above set is well-defined. Moreover, for any $m \in \mathcal{M}$,

$$\tilde{X}\frac{(Bm)_{1:d-1}}{(Bm)_d} - y = A\frac{Bm}{(Bm)_d} = \frac{Um}{(Bm)_d}.$$

Now let $\lambda \in \mathbb{R}^{d-1}$ with $\tilde{X}\lambda \neq y$, and define $v = DV^T[\lambda; 1] / \left\| \tilde{X}\lambda - y \right\|_2 \in \mathbb{R}^{d'}$. Observe that

$$\|v\|_2 = \|Uv\|_2 = \frac{\|A[\lambda; 1]\|_2}{\left\| \tilde{X}\lambda - y \right\|_2} = 1.$$

So there is some $m \in \mathcal{M}$ with $\|v - m\|_2 \leq \gamma$. Let $\lambda' = (Bm)_{1:d-1}/(Bm)_d \in \mathcal{N}$ and let $\sigma = \text{sign}((Bm)_d)$. Then

$$\frac{\tilde{X}\lambda - y}{\left\| \tilde{X}\lambda - y \right\|_2} = Uv$$

and

$$\frac{\tilde{X}\lambda' - y}{\left\| \tilde{X}\lambda' - y \right\|_2} = \frac{Um/(Bm)_d}{\|Um/(Bm)_d\|_2} = \text{sign}((Bm)_d) \cdot \frac{Um}{\|Um\|_2} = \sigma Um.$$

Thus,

$$\left\| \frac{\tilde{X}\lambda - y}{\left\| \tilde{X}\lambda - y \right\|_2} - \sigma \frac{\tilde{X}\lambda' - y}{\left\| \tilde{X}\lambda' - y \right\|_2} \right\|_2 = \|Uv - Um\|_2 = \|v - m\|_2 \leq \gamma$$

as desired. On the other hand, if $B_d$ is identically zero, then $V_d = 0$, so $y = (A^T)_d = 0$. This boundary case can be avoided by picking any nonzero covariate $(X^T)_i$ among $(X^T)_2, \ldots, (X^T)_d$ and replacing $y$ by $y + c(X^T)_i$ for a generic $c \in \mathbb{R}$; this does not change $\text{Stability}(X, y)$.

Note that a $\gamma$-net $\mathcal{M}$ of $\mathcal{S}^{d'-1}$ with $|\mathcal{M}| \leq (2\sqrt{d}/\gamma)^d$ can be constructed in time $O((\sqrt{d}/\gamma)^d)$ by discretizing the cube $[-1, 1]^{d'}$. Thus, $\mathcal{N}$ can be constructed in time $O((\sqrt{d}/\gamma)^d)$ as well, and $|\mathcal{N}| \leq |\mathcal{M}| \leq (2\sqrt{d}/\gamma)^d$. $\qquad\square$

All that is left is to show that under strong anti-concentration, the value of the linear program is Lipschitz in $\lambda$ (under the metric described in the previous lemma). This proves Theorem 1.5.

**Theorem E.2.** *Let $\epsilon, \delta > 0$. There is an algorithm* NETAPPROX *with time complexity* $(2\sqrt{d}/(\epsilon\delta^2))^d \cdot \text{poly}(n)$ *which, given arbitrary samples $(X_i, y_i)_{i=1}^n$, produces an estimate $\hat{S}$ satisfying* $\text{Stability}(X, y) \leq \hat{S}$. *Moreover, if $(X_i, y_i)_{i=1}^n$ satisfy $(\epsilon, \delta)$-strong anti-concentration, then in fact*

$$\text{Stability}(X, y) \leq \hat{S} \leq \text{Stability}(X, y) + 3\epsilon n + 1.$$

*Proof.* For any $\lambda \in \mathbb{R}^{d-1}$, define

$$V(\lambda) = \sup_{w \in [0,1]^n} \left\{ \|w\|_1 \Big| X^T(w \star (\tilde{X}\lambda - y)) = 0 \right\},$$

so that $n-\text{Stability}(X,y) = \sup_{\lambda \in \mathbb{R}^{d-1}} V(\lambda)$. For any fixed $\lambda$, we can compute $V(\lambda)$ in polynomial time, since it is defined as an LP with $n$ variables and $d$ constraints. Fix $\gamma = \epsilon\delta^2$. The algorithm NETAPPROX does the following (see Appendix J for pseudocode): if $X^T(\tilde{X}\lambda - y) = 0$ has a solution, then output 0. Otherwise, let $\mathcal{N}$ be the net guaranteed by Lemma E.1. Then compute $V(\lambda)$ for every $\lambda \in \mathcal{N}$, and output the estimate

$$\hat{S} = n - \max_{\lambda \in \mathcal{N}} V(\lambda).$$

Since the algorithm involves solving $(2\sqrt{d}/\gamma)^d$ linear programs, the time complexity is $(2\sqrt{d}/\gamma)^d \cdot \text{poly}(n)$ as claimed. Next, since the algorithm is maximizing $V(\lambda)$ over a subset of $\mathbb{R}^{d-1}$, it's clear that $\hat{S} \geq \text{Stability}(X,y)$. It remains to prove the upper bound on $\hat{S}$. Recall from Lemma B.1 that for any $\lambda \in \mathbb{R}^{d-1}$,

$$V(\lambda) = \inf_{u \in \mathbb{R}^d} \sup_{w \in [0,1]^n} \left\{ \|w\|_1 \left| \sum_{i=1}^n w_i(\langle \tilde{X}_i, \lambda \rangle - y_i)\langle X_i, u \rangle \geq 0 \right. \right\}.$$

Let $\lambda^*$ be a maximizer of $V(\lambda)$ and for notational convenience let $k = V(\lambda^*)$. If $\tilde{X}\lambda^* = y$, then the algorithm correctly outputs 0. Otherwise, by the guarantee of Lemma E.1, there is some $\lambda \in \mathcal{N}$ with $\tilde{X}\lambda \neq y$ and $\sigma \in \{-1, 1\}$ such that

$$\left\| \frac{\tilde{X}\lambda^* - y}{\left\|\tilde{X}\lambda^* - y\right\|_2} - \sigma \frac{\tilde{X}\lambda - y}{\left\|\tilde{X}\lambda - y\right\|_2} \right\|_2 \leq \gamma.$$

Pick any $u \in \mathbb{R}^d$. Since $V(\lambda^*) = k$, there is some $w^* = w^*(\sigma u) \in [0,1]^n$ such that $\|w^*\|_1 \geq k$ and

$$\sum_{i=1}^n w_i^*(\langle \tilde{X}_i, \lambda^* \rangle - y_i)\langle X_i, \sigma u \rangle \geq 0.$$

Without loss of generality, there is at most one coordinate $i \in [n]$ such that $w_i$ is strictly between 0 and 1. Also, the above inequality implies that

$$\sum_{i=1}^n w_i^* \frac{\langle \tilde{X}_i, \lambda \rangle - y_i}{\left\|\tilde{X}\lambda - y\right\|_2} \langle X_i, u \rangle \geq \sum_{i=1}^n w_i^* \left( \frac{\langle \tilde{X}_i, \lambda \rangle - y_i}{\left\|\tilde{X}\lambda - y\right\|_2} - \sigma \frac{\langle \tilde{X}_i, \lambda^* \rangle - y_i}{\left\|\tilde{X}\lambda^* - y\right\|_2} \right) \langle X_i, u \rangle$$

$$\geq - \left\| \frac{\tilde{X}\lambda - y}{\left\|\tilde{X}\lambda - y\right\|_2} - \sigma \frac{\tilde{X}\lambda^* - y}{\left\|\tilde{X}\lambda^* - y\right\|_2} \right\|_2 \|Xu\|_2$$

$$\geq -\gamma \|Xu\|_2$$

where the last two inequalities are by Cauchy-Schwarz and the guarantee of Lemma E.1, respectively. Now define $w \in [0,1]^n$ by the following procedure. Initially set $w := w^*$. Iterate through $[n]$ in increasing order of $(\langle \tilde{X}_i, \lambda \rangle - y_i)\langle X_i, u \rangle$ and repeatedly set the current coordinate $w_i := 0$, until

$$\sum_{i=1}^n w_i(\langle \tilde{X}_i, \lambda \rangle - y_i)\langle X_i, u \rangle \geq 0.$$

Obviously, this procedure will terminate with a feasible $w$. If it terminates after making $t$ updates, then $\|w\|_1 \geq \|w^*\|_1 - t$. Throughout the procedure, the sum $\sum_{i=1}^n w_i(\langle \tilde{X}_i, \lambda \rangle - y_i)\langle X_i, u \rangle$ is non-decreasing. If $|\langle \tilde{X}_i, \lambda \rangle - y_i| \geq (\delta/\sqrt{n})\left\|\tilde{X}\lambda - y\right\|_2$ and $|\langle X_i, u \rangle| \geq (\delta/\sqrt{n})\|Xu\|_2$ and $w_i^* = 1$, then the sum increases by at least $(\delta^2/n)\left\|\tilde{X}\lambda - y\right\|_2 \|Xu\|_2$. So the number of such steps is at most $\gamma n/\delta^2 \leq \epsilon n$. But the number of steps with $|\langle \tilde{X}_i, \lambda \rangle - y_i| < (\delta/\sqrt{n})\left\|\tilde{X}\lambda - y\right\|_2$ is at most $\epsilon n$ by $(\epsilon, \delta)$-strong anti-concentration. Similarly, the number of steps with $|\langle X_i, u \rangle| < (\delta/\sqrt{n})\|Xu\|_2$ is at most $\epsilon n$. And the number of steps with $0 < w_i^* < 1$ is at most 1. Thus, $\|w\|_1 \geq \|w^*\|_1 - 3\epsilon n - 1$. We conclude that $V(\lambda) \geq V(\lambda^*) - 3\epsilon n - 1$, so $\hat{S} \leq \text{Stability}(X,y) + 3\epsilon n + 1$. $\qquad\square$

# F   MOTIVATION FOR ANTI-CONCENTRATION ASSUMPTIONS

## F.1   SMOOTHING IMPLIES ANTI-CONCENTRATION

In this section, we show that anti-concentration (Assumption A) holds under the mild assumption that the response variable is *smoothed*. In the following proposition, notice that we always have the crude bound $\left\| X\beta^{(0)} - r \right\|_2 \leq \|r\|_2$.

**Proposition F.1.** *Let $\sigma > 0$ and let $(X_i, r_i)_{i=1}^n$ be arbitrary with $X_1, \ldots, X_n \in \mathbb{R}^d$ and $r_1, \ldots, r_n \in \mathbb{R}$. Suppose that $Z_1, \ldots, Z_n \sim N(0, \sigma^2)$ are independent Gaussian random variables, and define $y_i = r_i + Z_i$ for $i \in [n]$. Then with probability at least $1 - n^{-d} - 2e^{-n}$, the dataset $(X_i, y_i)_{i=1}^n$ satisfies $(2d/n, \delta)$-anti-concentration where*

$$\delta = \min\left( \frac{\sigma}{n\sqrt{d}\left\|X\beta^{(0)} - r\right\|_2}, \frac{1}{3n^3\sqrt{2d}} \right)$$

*and $\beta^{(0)} \in OLS(X, r, \mathbb{1})$.*

*Proof.* Without loss of generality, $\beta^{(0)} = (X^T X)^\dagger X^T r$. Define $\hat{\beta} = (X^T X)^\dagger X^T y$; we want to upper bound $\left\| X\hat{\beta} - y \right\|_2$. Let $Q = X(X^T X)^\dagger X^T$; then $y - X\hat{\beta} \sim N((I - Q)r, \sigma^2(I - Q))$. But $(I - Q)r = r - X\beta^{(0)}$, and $N(0, \sigma^2(I - Q))$ is stochastically dominated by $N(0, \sigma^2 I)$. So $\left\| y - X\hat{\beta} \right\|_2 \leq \left\| r - X\beta^{(0)} \right\|_2 + 2\sigma\sqrt{n}$ with probability at least $1 - 2e^{-n}$, by the tail bound for $\chi^2$ random variables.

Next, we claim that with high probability, for every set $S \subseteq [n]$ of size $|S| = 2d$, there is no $\beta \in \mathbb{R}^d$ such that $|X_i\beta - y_i| \leq \sigma/(n^3\sqrt{2d})$ for all $i \in [S]$. Fix some $S$. We bound the probability that for all $\beta \in \mathbb{R}^d$,

$$\sum_{i \in S}(X_i\beta - y_i)^2 \leq \frac{\sigma^2}{n^6}$$

because this event contains the event that all residuals in $S$ are at most $\sigma/(n\sqrt{2d})$. Now it suffices to restrict to the OLS estimator $\hat{\beta}_{\text{OLS}} = (X_S^T X_S)^\dagger X_S^T y_S$. Now defining $P = X_S(X_S^T X_S)^\dagger X_S^T$, we have

$$\begin{aligned}
y_S - X_S\hat{\beta}_{\text{OLS}} &= (r_S + Z_S) - P(r_S + Z_S) \\
&= (I - P)(r_S + Z_S) \\
&\sim N((I - P)r_S, \sigma^2(I - P)).
\end{aligned}$$

Note that $I - P$ is an orthogonal projection onto a space of dimension $d' := |S| - \text{rank}(X_S) \geq d$, so there is a matrix $M \in \mathbb{R}^{2d \times d'}$ be such that $MM^T = I - P$ and $M^T M = I_{d'}$. Then we can write

$$\begin{aligned}
y_S - X_S\hat{\beta}_{\text{OLS}} &= \sigma MA + (I - P)r_S \\
&= M(\sigma A + M^T r_S)
\end{aligned}$$

for a random vector $A \sim N(0, I_{d'})$. This means that $\left\| y_S - X_S\hat{\beta}_{\text{OLS}} \right\|_2^2 = \left\| \sigma A + M^T r_S \right\|_2^2$. But for any $\mu \in \mathbb{R}$, note that $|\xi + \mu|$ stochastically dominates $|\xi|$ where $\xi \sim N(0, 1)$, so $(\xi + \mu)^2$ stochastically dominates $\xi^2$, and thus $\left\| \sigma A + M^T r_S \right\|_2^2$ stochastically dominates $\left\| \sigma A \right\|_2^2$. Therefore

$$\Pr\left[ \left\| y_S - X_S\hat{\beta}_{\text{OLS}} \right\|_2^2 \leq \frac{\sigma^2}{n^6} \right] \leq \Pr\left[ A_1^2 + \cdots + A_{d'}^2 \leq \frac{1}{n^6} \right].$$

For any $1 \leq i \leq d'$, since the density of $A_i$ is bounded above by $1/\sqrt{2\pi}$, we have

$$\Pr[A_1^2 + \cdots + A_d^2 \leq 1/n^6] \leq \prod_{i=1}^{d'} \Pr[|A_i| \leq 1/n^3] \leq \left( \frac{\sqrt{2}}{n^3\sqrt{\pi}} \right)^{d'} \leq n^{-3d'}.$$

Recalling that $d' \geq d$, this shows that for a fixed $S \subseteq [n]$ of size $2d$, with probability at least $1 - n^{-3d}$, there is no $\beta \in \mathbb{R}^d$ with $|X_i\beta - y_i| \leq \sigma/(n^3\sqrt{2d})$ for all $i \in S$. A union bound over sets $S$ of size $2d$ proves the claim.

Finally, we use the above bounds to show that the smoothed data satisfies $(2d/n, \delta)$-anti-concentration. We consider two cases. First, if $\left\|X\beta^{(0)} - r\right\|_2 \leq \sigma\sqrt{n}$, then with probability $1 - 2e^{-n}$ we have $\left\|X\hat{\beta} - y\right\|_2 \leq 3\sigma\sqrt{n}$, so for any $\beta \in \mathbb{R}^d$, the number of $i \in [n]$ satisfying

$$|\langle X_i, \beta\rangle - y_i| \leq \frac{\delta}{\sqrt{n}}\left\|X\hat{\beta} - y\right\|_2 \leq \frac{1}{3n^3\sqrt{2nd}} \cdot \left\|X\hat{\beta} - y\right\|_2 \leq \frac{\sigma}{n^3\sqrt{2d}}$$

is at most $\epsilon n$. $\qquad\square$

## F.2 Distributional Assumptions for strong anti-concentration

In this section, we show that under reasonable distributional assumptions on the samples $(X_i, y_i)_{i=1}^n$, strong anti-concentration (Assumption B) holds with constant $\epsilon, \delta > 0$. First, it holds if the samples $(X_i, y_i)$ are i.i.d. and have an arbitrary multivariate Gaussian joint distribution.

**Proposition F.2.** *Let $n, d, \epsilon > 0$ and set $\delta = \epsilon/4$. Let $\Sigma : d \times d$ be symmetric and positive-definite. Let $Z_1, \ldots, Z_n \sim N(0, \Sigma)$ be independent and identically distributed. If $n \geq Cd\log(n)/\epsilon^2$ for some absolute constant $C$, then with probability at least $1 - \exp(-\Omega(\epsilon^2 n))$, samples $(Z_i)_{i=1}^n$ satisfy $(\epsilon, \delta)$-strong anti-concentration, i.e. for all $\beta \in \mathbb{R}^d$, it holds that*

$$\left|\left\{i \in [n] : |\langle Z_i, \beta\rangle| < \frac{\delta}{\sqrt{n}}\|Z\beta\|_2\right\}\right| \leq \epsilon n$$

*where $Z : n \times d$ is the matrix with rows $Z_1, \ldots, Z_n$.*

*Proof.* Let $\hat{\Sigma} = \frac{1}{n}\sum_{i=1}^n Z_i Z_i^T$. Then by concentration of Wishart matrices, it holds with probability at least $1 - 2\exp(-(n-d)/2)$ that $\hat{\Sigma} \preceq 2\Sigma$ (see e.g. Exercise 4.7.3 in Vershynin (2018)). In this event, $\|Z\beta\|_2^2/n \leq 2\mathbb{E}|\langle Z_1, \beta\rangle|^2$ for all $\beta \in \mathbb{R}^d$.

Let $\mathcal{F} = \{f_\beta : \beta \in \mathbb{R}^d\}$ be the class of binary functions $f_\beta(x) = \mathbb{1}[|\langle x, \beta\rangle|^2 < (\epsilon^2/4)\mathbb{E}|\langle Z_1, \beta\rangle|^2]$. Observe that every function in $\mathcal{F}$ is the intersection of parallel half-spaces, so $\mathcal{F}$ has VC dimension $O(d)$. Moreover, for any $\beta$,

$$\mathbb{E}f_\beta(Z_1) = \Pr[|\langle Z_1, \beta\rangle|^2 < (\epsilon^2/4)\beta^T\Sigma\beta] \leq \frac{\epsilon}{2}$$

since $\langle Z_1, \beta\rangle \sim N(0, \beta^T\Sigma\beta)$, and Gaussian random variables are anti-concentrated ($\Pr(|\xi| < c) \leq c$ for any $c > 0$ and $\xi \sim N(0, 1)$). Thus, by the Vapnik-Chervonenkis bound and assumption on $n$,

$$\Pr\left(\sup_{f \in \mathcal{F}} \frac{1}{n}\sum_{i=1}^n f_\beta(Z_i) > \mathbb{E}f_\beta(Z_1) + \epsilon/2\right) \leq \exp(O(d\log n) - \Omega(\epsilon^2 n)) \leq \exp(-\Omega(\epsilon^2 n)).$$

So with probability at least $1 - \exp(-\Omega(\epsilon^2 n))$ it holds that for all $\beta \in \mathbb{R}^d$,

$$\left|\left\{i \in [n] : |\langle Z_i, \beta\rangle|^2 < (\epsilon^2/4)\beta^T\Sigma\beta\right\}\right| \leq \epsilon n.$$

This means that with probability at least $1 - \exp(-\Omega(\epsilon^2 n)) - \exp(-(n-d)/2)$, we have that for all $\beta \in \mathbb{R}^d$,

$$\left|\left\{i \in [n] : |\langle Z_i, \beta\rangle|^2 < (\epsilon^2/8)\|Z\beta\|_2^2/n\right\}\right| \leq \epsilon n$$

as desired. $\qquad\square$

Second, strong anti-concentration holds if the samples are drawn from a mixture of centered Gaussian distributions, with arbitrary weights, so long as each covariance matrix has bounded largest and smallest eigenvalues.

**Proposition F.3.** *Let $n, d, k, \epsilon > 0$ and let $\Sigma_1, \ldots, \Sigma_k$ be symmetric and positive-definite matrices. Suppose that there are constants $\lambda, \Lambda > 0$ such that $\lambda I \preceq \Sigma_1, \ldots, \Sigma_k \preceq \Lambda I$, and define $\kappa = \Lambda / \lambda$. For arbitrary weights $w_1, \ldots, w_k \geq 0$ with $w_1 + \cdots + w_k = 1$, let $Z_1, \ldots, Z_n \sim \sum_{i=1}^k w_i N(0, \Sigma_i)$ be independent and identically distributed. If $n \geq Cd \log(n)/\epsilon^2$ for some absolute constant $C$, then with probability at least $1 - \exp(-\Omega(\epsilon^2 n))$, samples $(Z_i)_{i=1}^n$ satisfy $(\epsilon, \epsilon/(4\kappa))$-strong anti-concentration.*

*Proof.* The proof is similar to that of the previous proposition. First, note that each $Z_i$ can be coupled with $\xi \sim N(0, \Lambda I)$ so that $Z_i Z_i^T \preceq \xi \xi^T$. So with probability at least $1 - 2\exp(-(n - d)/2)$ it holds that $\frac{1}{n} \sum_{i=1}^n Z_i Z_i^T \preceq 2\lambda I$ and thus $\|Z\beta\|_2^2/n \leq 2\Lambda \|\beta\|_2^2$ for all $\beta \in \mathbb{R}^d$.

Next, let $\mathcal{F} = \{f_\beta : \beta \in \mathbb{R}^d\}$ be the class of binary functions $f_\beta(x) = \mathbb{1}[|\langle x, \beta \rangle|^2 < \epsilon^2 \lambda \|\beta\|_2^2 /4]$. Then $\mathcal{F}$ has VC dimension $O(d)$, and for any $\beta$,

$$\mathbb{E}f_\beta(Z_1) = \Pr[|\langle Z_1, \beta \rangle|^2 < \epsilon^2 \lambda \|\beta\|_2^2 /4] \leq \frac{\epsilon}{2}$$

since $\langle Z_1, \beta \rangle \sim \sum_{i=1}^k w_i N(0, \beta^T \Sigma_i \beta)$ is stochastically dominated by $N(0, \lambda \|\beta\|_2^2)$. By the Vapnik-Chervonenkis bound, with probability at least $1 - \exp(-\Omega(\epsilon^2 n))$, we get that for all $\beta \in \mathbb{R}^d$,

$$|\{i \in [n] : |\langle Z_i, \beta \rangle|^2 < (\epsilon^2/4)\lambda \|\beta\|_2^2\}| \leq \epsilon n.$$

Combining with the upper bound on $\|Z\beta\|_2^2$, we have with probability at least $1 - \exp(-\Omega(\epsilon^2 n)) - \exp(-(n - d)/2)$ that for all $\beta \in \mathbb{R}^d$,

$$|\{i \in [n] : |\langle Z_i, \beta \rangle|^2 < (\epsilon^2/(8\kappa)) \|Z\beta\|_2^2 /n\}| \leq \epsilon n$$

as claimed. $\square$

# G    EXTENSION TO IV LINEAR REGRESSION

We extend the definition of stability to measure the stability of the sign of a coefficient of the IV linear regressor:

**Definition G.1.** For samples $(X_i, y_i, Z_i)_{i=1}^n$ with covariates $X_i \in \mathbb{R}^d$, response $y_i \in \mathbb{R}^d$, and instruments $Z_i \in \mathbb{R}^p$, the ordinary IV estimator set with weight vector $w \in [0, 1]^n$ is

$$\mathsf{IV}(X, y, Z, w) = \{\beta \in \mathbb{R}^d : Z^T(w \star (X\beta - y)) = 0\}.$$

The finite-sample stability of $(X_i, y_i, Z_i)_{i=1}^n$ is then defined

$$\text{IV-Stability}(X, y, Z) := \inf_{w \in [0,1]^n, \beta \in \mathbb{R}^d} \{n - \|w\|_1 : \beta_1 = 0 \wedge \beta \in \mathsf{IV}(X, y, Z, w)\}.$$

For any $k$, the expression IV-Stability$(X, y, Z) \geq k$ is still defined as a bilinear system of equations in $\beta$ and $w$. Our exact algorithm for Stability$(X, y)$ never uses that the weighted residual $(w \star (X\beta - y))$ is multiplied by $X^T$ in the OLS solution set, rather than some arbitrary matrix $Z^T$; all that matters is that this matrix has at most $d$ rows. Thus, with a bound on the number of instruments, the algorithm generalizes to computing IV-Stability$(X, y, Z)$:

**Theorem G.2.** *There is an $n^{O(dp(d+p))}$-time algorithm which, given $n$ arbitrary samples $(X_i, y_i, Z_i)_{i=1}^n$ with $X_1, \ldots, X_n \in \mathbb{R}^d$ and $Z_1, \ldots, Z_n \in \mathbb{R}^p$ and $y_1, \ldots, y_n \in \mathbb{R}$, and given $k \geq 0$, decides whether IV-Stability$(X, y, Z) \leq k$.*

The algorithm NETAPPROX also generalizes to IV regression. To state the guarantee, we define an anti-concentration assumption for IV regression data.

**Assumption C.** *Let $\epsilon, \delta > 0$. We say that samples $(X_i, y_i, Z_i)_{i=1}^n$ satisfy $(\epsilon, \delta)$-strong anti-concentration if for every $\alpha \in \mathbb{R}^p$ and $\beta \in \mathbb{R}^d$ it holds that*

$$|\{i \in [n] : |\langle Z_i, \alpha \rangle| < (\delta/\sqrt{n}) \|Z\alpha\|_2\}| \leq \epsilon n$$

*and*

$$|\{i \in [n] : |\langle \tilde{X}_i, \beta \rangle - y_i| < (\delta/\sqrt{n}) \|\tilde{X}\beta - y\|_2\}| \leq \epsilon n.$$

Then it is easy to see that the proof of Theorem 1.5 immediately extends to give the following result:

**Theorem G.3.** *For any $\epsilon, \delta > 0$, there is a $(\sqrt{d}/(\epsilon\delta^2))^d \cdot \text{poly}(n)$-time algorithm which, given $\epsilon, \delta$, and samples $(X_i, y_i, Z_i)_{i=1}^n$ satisfying $(\epsilon, \delta)$-strong anti-concentration, returns an estimate $\hat{S}$ satisfying*

$$\text{IV-Stability}(X, y, Z) \leq \hat{S} \leq \text{IV-Stability}(X, y, Z) + 3\epsilon n + 1.$$

## H    HEURISTIC FOR LOWER BOUNDING STABILITY

In this section we explain the "LP lower bound" which we applied in Section 5 to provide exact lower bounds on stability of various datasets.

Given a list of thresholds $T$ and a subset size $m$, we randomly pick a set $S \subseteq [n]$ of size $m$, and enumerate the regions $R_1, \ldots, R_p$ defined by the hyperplanes

$$\langle \tilde{X}_i, \lambda \rangle - y_i = t \qquad \forall i \in S, \forall t \in T.$$

Fix one such region $R$. Similar to PARTITIONANDAPPROX, we use the change of variables $g_i = (1 - w_i)(\langle \tilde{X}_i, \lambda \rangle - y_i)$, and enforce the constraint $X^T g = X^T (\tilde{X}\lambda - y)$. For some of the $i \in [n]$, the residual $\langle \tilde{X}_i, \lambda \rangle - y_i$ will have constant sign on the entire region. For these samples, we enforce the constraint $0 \leq g_i \leq \langle \tilde{X}_i, \lambda \rangle - y_i$ (if the residual is non-negative) or $\langle \tilde{X}_i, \lambda \rangle - y_i \leq g_i \leq 0$ (if the residual is non-positive). However, because we didn't include a hyperplane for every sample, it's likely that for some $i \in [n]$, the residual attains both signs within the region, in which case the constraint $w_i \in [0, 1]$ is not convex in $g$ and $\lambda$. Thus, we relax the constraint to

$$\inf_{\lambda \in R} \langle \tilde{X}_i, \lambda \rangle - y_i \leq g_i \leq \sup_{\lambda \in R} \langle \tilde{X}_i, \lambda \rangle - y_i.$$

Note that this is indeed a relaxation, because the interval contains 0. Finally, let $K_+ \subseteq [n]$ be the set of indices for which the residual is non-negative on $R$, and let $K_- \subseteq [n]$ be the set of indices for which the residual is non-positive. Then we minimize the objective

$$\sum_{i \in K_+} \frac{g_i}{\sup_{\lambda \in R} \langle \tilde{X}_i, \lambda \rangle - y_i} + \sum_{i \in K_-} \frac{g_i}{\inf_{\lambda \in R} \langle \tilde{X}_i, \lambda \rangle - y_i}.$$

Because this objective is less than or equal to $\sum_{i \in [n]} g_i / (\langle \tilde{X}_i, \lambda \rangle - y_i)$, and because we only relaxed constraints, this program has value at most $V_R$ (the value of the exact non-linear program restricted to $\lambda \in R$).

Compared to the provable approximation algorithm described previously, this algorithm is a heuristic because each region will typically have samples for which the residual changes sign within the region. As rough intuition, if the residual $\langle \tilde{X}_i, \lambda \rangle - y_i$ remains fairly small throughout the region then the relaxation of the constraint on $w_i$ may not be problematic. However, if the residual can attain large magnitude in the region, then relaxing the constraint on $w_i$ may significantly change the value of the program. However, we may expect that if the partition is sufficiently fine, then a region where some residual is allowed to blow up may have many samples whose residuals are forced to be large. This motivates the use of an additional heuristic to refine the certification algorithm: if $(w^*, \beta^*)$ are the optimal solution to the sensitivity problem, and $\beta^{(0)} = \text{OLS}(X, y, \mathbb{1})$ is the original regressor, then it must hold that

$$\sum_{i=1}^n w_i^* (\langle X_i, \beta^* \rangle - y_i)^2 \leq \sum_{i=1}^n w_i^* (\langle X_i, \beta^{(0)} \rangle - y_i)^2 \leq \sum_{i=1}^n (\langle X_i, \beta^{(0)} \rangle - y_i)^2.$$

However, if $R$ is the region containing the optimal solution and if $\|w^*\|_1 \geq n - k$, then

$$\sum_{i=1}^n w_i^* (\langle X_i, \beta^* \rangle - y_i)^2 \geq \inf_{S \subseteq [n]: |S| = n-k} \sum_{i \in S} (\langle X_i, \beta^* \rangle - y_i)^2 \geq \inf_{S \subseteq [n]: |S| = n-k} \sum_{i \in S} \inf_{\lambda \in R} (\langle \tilde{X}_i, \lambda \rangle - y_i)^2.$$

Thus, if we compute $Q_i(R) = \inf_{\lambda \in R} (\langle \tilde{X}_i, \lambda \rangle - y_i)^2$ (by computing the interval of achievable residuals) for each $i \in [n]$, then we can lower bound $k$ (conditioned on $R$ containing the optimal solution) by sorting $Q_1(R), \ldots, Q_n(R)$ and finding the smallest subset $K \subseteq [n]$ such that $\sum_{i \notin K} Q_i(R) \leq \|X\beta^{(0)} - y\|_2^2$.

To summarize the algorithm, for each region we compute the maximum of the LP lower bound and the residual lower bound, and the output of the algorithm is the minimum over all regions. See Algorithm 4 in Appendix J for detailed pseudocode.

## I FURTHER EXPERIMENTAL DETAILS

### I.1 IMPLEMENTED ALGORITHMS

**Net upper bound.** We implement NETAPPROX with one modification: instead of deterministically picking the net $\mathcal{M}$ by discretization (see Lemma E.1), we let $\mathcal{M}$ be a set of random unit vectors from $\mathcal{S}^{d'-1}$, and then compute $\mathcal{N}$ as in Lemma E.1. Instead of parametrizing the algorithm by the desired approximation error $\epsilon$, we parametrize by $|\mathcal{M}|$. Despite this change, the algorithm still provides a provable, exact *upper bound* on $\mathrm{Stability}(X, y)$.

**LP lower bound.** The disadvantage of NETAPPROX is that it only lower bounds the stability under an assumption that seems hard to check. The PARTITIONANDAPPROX algorithm is better, because it unconditionally, with high probability, outputs either an accurate estimate or a failure symbol $\perp$. However, the $\Omega(n^d)$ time complexity (needed so that in each region, all $n$ residuals have constant sign) may be prohibitively slow in practice. For this reason, we introduce a heuristic simplification of PARTITIONANDAPPROX which provably *lower bounds* the stability with no assumptions.

At a high level, we decrease the number of regions by ignoring the requirement that within each region all residuals have constant sign. The algorithm is parametrized by a list of thresholds $L$ and a subset size $m$, and the regions are demarcated by the hyperplanes $\langle \tilde{X}_i, \lambda \rangle - y_i = t$ for $m$ random choices $i \in [n]$ and all $t \in L$. Now, for samples which do not have constant residual in a particular region $R$, the constraints $w_i \in [0, 1]$ are nonconvex after the change of variables. We relax these constraints to linear constraints, and relax the objective function to skip the "bad" samples. Heuristically, this relaxation should not lose too much on samples where the residual remains small throughout $R$, but may be problematic if the residual blows up. This motivates the use of a complementary lower bound heuristic based on the minimum squared-loss achievable by any $\lambda \in R$. See Appendix H for details.

**Baseline greedy upper bound (Broderick et al., 2020; Kuschnig et al., 2021).** We implement the greedy algorithm described by Kuschnig et al. (2021), which refines the algorithm of Broderick et al. (2020): iteratively remove the sample with the largest local influence until the sign of the first coefficient of the OLS is reversed. After each step, recompute the influences.

**Baseline lower bound.** This algorithm simply computes the squared-loss-based lower bound (used in our full lower bound algorithm) for each region. See Appendix H for details.

### I.2 HYPERPARAMETER CHOICES

The net upper bound has only one hyperparameter (the number of trials), which should be chosen as large as possible subject to computational constraints. The LP lower bound has two hyperparameters (the size of the random subsample, and the set of thresholds). We choose these ad hoc subject to our computational constraints. Experiments to determine the optimal tradeoff between the subsample size and threshold set could be useful. However, we note that because the LP lower bound *unconditionally* lower bounds the stability, no matter what hyperparameters we choose, in practice it suffices to try several sets of hyperparameters and compute the maximum of the resulting lower bounds.

**Heterogeneous data experiment.** For each dataset, we applied the net upper bound with 1000 trials, the LP lower bound with $L = \{-0.01, 0, 0.01\}$ and $m = 30$, and the baseline lower bound with $L = \{-0.01, 0, 0.01\}$ and $m = 1000$.

**Isotropic Gaussian data.** For each dataset with noise level $\sigma$, we applied the net upper bound with $10^d$ trials, the LP lower bound with $L = \{-\sigma, 0, \sigma\}$ and $m = 30$, and the baseline lower bound with $L = \{-\sigma, 0, \sigma\}$ and $m = 1000$.

**Boston Housing dataset.** For the two-dimensional datasets, we applied the net upper bound with 100 trials and the LP lower bound with $L = \{0\}$ and $m = 100$. For the three-dimensional dataset we applied the net upper bound with 1000 trials and the LP lower bound with $L = \{0\}$ and $m = 30$.

### I.3 INDEPENDENT TRIALS & ERROR BARS

For randomized algorithms with two-sided errors, standard practice is to run the algorithm multiple times on the same dataset and report e.g. the median and error bars. However, our algorithms provide unconditional upper/lower bounds on the stability, so this is not necessary. If we did run the LP lower bound multiple times, we could simply report the maximum outcome, and this would be a valid lower bound on the stability; similarly for the net upper bound we could report the minimum outcome.

However, for our synthetic two-dimensional datasets (Figures 1 and 2a), because the data itself is random, we construct each dataset 10 independent times. For each algorithm and each dataset we compute the stability bound. Then, we report the median bound and error bars across the 10 independent-but-identically-constructed datasets. All error bars are 25th and 75th percentiles.

There are no error bars in our synthetic three-dimensional experiment (Figure 2b) because we only conducted one trial per noise level (rather than 10) due to computational constraints.

### I.4 COMPUTATIONAL DETAILS

All experiments were done in Python on a Microsoft Surface Laptop, using GUROBI (Gurobi Optimization, LLC, 2022) with an Academic License to solve the linear programs. Each plot took at most 30 hours to generate (specifically, the three-dimensional Gaussian isotropic data experiment took 3 hours for each of the 10 datasets, dominated by the time required for the LP lower bound algorithm; other plots were faster).

### I.5 OMITTED EXPERIMENT: COVARIANCE SHIFT

Consider a dataset with $n$ samples $(X_i, y_i)$ drawn from $X_i \sim N(0, \Sigma)$ and $y_i = -X_{i1} + X_{i2}$, where $\Sigma = \begin{bmatrix} 1 & -1 \\ -1 & 2 \end{bmatrix}$. Additionally, there are $k + 1$ outliers, in two types: the $k$ type-I outliers $(X_i, y_i)$ have $X_i = c(1, -3)$ and $y_i = -C$ large and negative; the one type-II outlier $(X_i, y_i)$ has $X_i = \sqrt{n}(1, 1)$ and $y_i$ chosen to exactly lie on the OLS best-fit hyperplane $y = \langle x, \hat{\beta} \rangle$. Initially, $\hat{\beta}_1 > 0$, and clearly removing the last $k + 1$ samples suffices to flip the sign. However, the initial sample covariance is roughly $\hat{\Sigma} = \Sigma + \mathbb{1}\mathbb{1}^T = \begin{bmatrix} 2 & 0 \\ 0 & 3 \end{bmatrix}$. The influence of a sample $(X_i, y_i)$ on coordinate $j$ of the OLS regressor $\hat{\beta}$ is $\langle (\hat{\Sigma}^{-1})_j, X_i \rangle \cdot (y_i - \langle X_i, \hat{\beta} \rangle)$. As a result, the type-I outliers have *negative* influence on $\hat{\beta}_1$, so the greedy algorithm initially does not remove them. The influence only becomes positive after removing the type-II outlier, because this shifts the sample covariance to $\Sigma$, and therefore flips the sign of $\langle (\hat{\Sigma}^{-1})_1, X_i \rangle$.

Constructing this example experimentally requires some care in the choices of $k$, $c$, and $C$: we need the total influence of type-A outliers (proportional to $kcC$) large enough that $\hat{\beta}_1 > 0$, and need $kc^2$ small enough that the type-I outliers don't affect the sample covariance much. Moreover, the number of trials needed by the net algorithm roughly scales with $C$. We take $n = 1000$, $k = 30$, $c = 0.2$, and $C = 300$. Applying the greedy baseline and the net upper bound (with 1000 trials), we find that the former removes 97 samples while the latter removes roughly 16.7 samples. In this example, the failure of the greedy baseline can be attributed to the constant-factor shift in the sample covariance achieved by removing the type-B outlier.

### I.6 OMITTED FIGURES

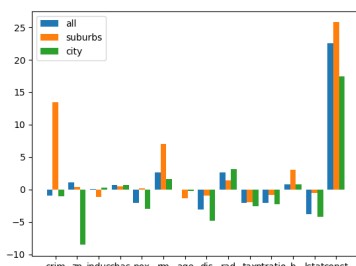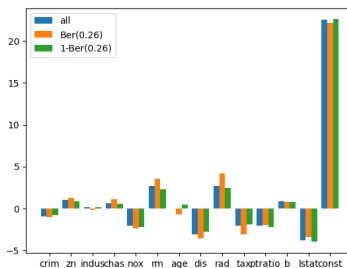

Figure 4: Coefficients of OLS regression for the suburb/city split (Figure (a)) and a random split with the same ratio (Figure (b)). In both cases, we first rescale all covariates to mean $0$ and variance $1$, and add a constant variable (the last coefficient in both plots). In Figure (a), we then plot the OLS regressor on all samples (in blue), the OLS regressor on all $134$ zn $> 0$ samples (in orange), and the OLS regressor on all $472$ zn $= 0$ samples (in green). In Figure (b), we pick a subset $S$ of samples of expected size $134$. We plot the OLS regressor on all samples (in blue), the OLS regressor on $S$ (in orange), and the OLS regressor on $S^c$ (in green).

## J FORMAL PSEUDOCODE FOR ALGORITHMS

---
**Algorithm 1** NETAPPROX
---
1: **procedure** NETAPPROX($(X_i, y_i)_{i=1}^n, \epsilon, \delta$)
2:  $\quad \tilde{X} \leftarrow [(X^T)_2; \ldots, (X^T)_d]$
3:  $\quad$ **if** $X^T(\tilde{X}\lambda - y) = 0$ has a solution $\lambda$ **then**
4:  $\quad\quad$ **return** $0$
5:  $\quad \gamma \leftarrow \epsilon\delta^2$
6:  $\quad A \leftarrow [(X^T)_2; \ldots; (X^T)_d; -y], d' \leftarrow \text{rank}(A)$
7:  $\quad U, D, V \leftarrow \mathsf{SVD}(A)$ (so that $A = UDV^T$ and $D$ is diagonal matrix of the $d'$ nonzero singular values)
8:  $\quad B \leftarrow VD^{-1}$
9:  $\quad$ Define $\gamma$-net for $\mathcal{S}^{d'-1}$ by

$$\mathcal{M} \leftarrow \{R(\pm m_1, \pm m_2, \ldots, \pm m_{d'}) : m_1, \ldots, m_{d'} \in \{0, \gamma/\sqrt{d'}, 2\gamma/\sqrt{d'}, \ldots, 1\}\}$$

$\quad$ where $R : d' \times d'$ is uniformly random rotation matrix
10:  $\quad$ Define $\gamma$-net for "residual space" (Lemma E.1) by

$$\mathcal{N} \leftarrow \left\{\frac{(Bm)_{1:d-1}}{(Bm)_d} : m \in \mathcal{M}\right\}.$$

11:  $\quad \hat{S} \leftarrow n$
12:  $\quad$ **for** $\lambda \in \mathcal{N}$ **do**
13:  $\quad\quad$ Solve linear program

$$V(\lambda) = \sup_{w \in [0,1]^n} \left\{\|w\|_1 \Big| X^T(w \star (\tilde{X}\lambda - y)) = 0\right\}.$$

14:  $\quad\quad \hat{S} \leftarrow \min(\hat{S}, n - V(\lambda))$
15:  $\quad$ **return** $\hat{S}$

---

---

**Algorithm 2** GENERATEREGIONS

---

1: **procedure** GENERATEREGIONS($(v_i, c_i)_{i=1}^N$)
2:     $\mathcal{R} \leftarrow \{\mathbb{R}^d\}$                          $\triangleright$ List of regions, each described by linear constraints
3:     **for** $(v_i, t_i)$ **do**
4:         $\mathcal{R}_{\text{next}} \leftarrow \mathcal{R}$
5:         **for** $R \in L$ **do**
6:             Solve linear programs

$$c_{\text{low}} = \inf_{x \in R} \langle v_i, x \rangle \qquad \text{and} \qquad c_{\text{high}} = \sup_{x \in R} \langle v_i, x \rangle$$

7:             **if** $c_{\text{low}} < c < c_{\text{high}}$ **then**
8:                 Add regions $R \cap \{\langle v_i, x \rangle \leq c_i\}$ and $R \cap \{\langle v_i, x \rangle \geq c_i\}$ to $\mathcal{R}_{\text{new}}$
9:             **else**
10:                 Add region $R$ to $\mathcal{R}_{\text{new}}$
11:         $\mathcal{R} \leftarrow \mathcal{R}_{\text{new}}$
12:     **return** $\mathcal{R}$

---

---

**Algorithm 3** PARTITIONANDAPPROX

---

1: **procedure** PARTITIONANDAPPROX($(X_i, y_i)_{i=1}^n, \delta, \epsilon, \eta$)
2:      $\beta^{(0)} \leftarrow \arg\min_{\beta \in \mathbb{R}^d} \|X\beta - y\|_2^2$          $\triangleright$ $X : n \times d$ matrix with rows $X_1, \ldots, X_n$
3:      $M \leftarrow \|X\beta^{(0)} - y\|_2$
4:      $\tilde{X} \leftarrow [(X^T)_2; \ldots; (X^T)_d]$
5:      $m \leftarrow C\epsilon^{-1}(d \log \epsilon^{-1} + \log \eta^{-1})$          $\triangleright$ $C > 0$ is a universal constant
6:      Sample $i_1, \ldots, i_m \in [n]$ without replacement
7:      $\mathcal{H} \leftarrow \{\}$
8:      **for** $i \in [n]$ **do**
9:          Add $(\tilde{X}_i, y_i)$ to $\mathcal{H}$          $\triangleright$ Describes hyperplane $\langle \tilde{X}, \lambda \rangle = y_i$
10:          Add $(\tilde{X}_i, y_i + M), (\tilde{X}_i, y_i - M), (\tilde{X}_i, y_i + \delta M/\sqrt{n})$ and $(\tilde{X}_i, y_i - \delta M/\sqrt{n})$ to $\mathcal{H}$
11:      **for** $j \in [m]$ **do**
12:          **for** $0 \leq k \leq \lceil \log_{1+\epsilon}(1/\delta) \rceil$ **do**
13:              Add $(\tilde{X}_{i_j}, y_{i_j} + (1+\epsilon)^k \delta M)$ and $(\tilde{X}_{i_j}, y_{i_j} - (1+\epsilon)^k \delta M)$ to $\mathcal{H}$
14:      $\mathcal{R} \leftarrow$ GENERATEREGIONS($\mathcal{H}$)
15:      **for** $R \in \mathcal{R}$ **do**
16:          Find feasible point $\lambda_0(R) \in R$
17:          $B_M \leftarrow \{i \in [n] : |\langle \tilde{X}_i, \lambda_0(R) \rangle - y_i| > M\}$
18:          $B_{\delta M} \leftarrow \{i \in [n] : |\langle \tilde{X}_i, \lambda_0(R) \rangle - y_i| < \delta M/\sqrt{n}\}$
19:          **if** $|B_{\delta M}| > \epsilon n$ **then**
20:              **return** $\perp$
21:          **for** $i \in [n]$ **do**
22:              **if** $\inf_{\lambda \in R} \langle \tilde{X}_i, \lambda \rangle - y_i \geq 0$ **then**
23:                  $\sigma_i \leftarrow 1$
24:              **else**
25:                  $\sigma_i \leftarrow -1$
26:          Solve linear program

$$(\hat{g}(R), \hat{\lambda}(R))$$

$$\leftarrow \arg\sup_{g \in \mathbb{R}^n, \lambda \in R, s \in \mathbb{R}^n} \left\{ \sum_{i \in [n] \setminus (B_M \cup B_{\delta M})} s_i \left| \begin{array}{ll} X^T g = 0 & \\ 0 \leq g_i \leq \langle \tilde{X}_i, \lambda \rangle - y_i & \forall i \in [n] : \sigma_i = 1 \\ \langle \tilde{X}_i, \lambda \rangle - y_i \leq g_i \leq 0 & \forall i \in [n] : \sigma_i = -1 \\ s_i \leq 1 & \forall i \in [n] \setminus (B_M \cup B_{\delta M}) \\ s_i \leq g_i/(\langle \tilde{X}_i, \lambda_0(R) \rangle - y_i) & \forall i \in [n] \setminus (B_M \cup B_{\delta M}) \end{array} \right. \right\}$$

27:      **return**

$$\max_{R \in \mathcal{R}} \sum_{i \in [n]} \frac{\hat{g}(R)_i}{\langle \tilde{X}_i, \hat{\lambda}(R) \rangle - y_i}.$$

---

---

**Algorithm 4** LP Lower Bound

---

1: **procedure** LPLOWERBOUND($(X_i, y_i)_{i=1}^n, L, m$)
2: $\quad \beta^{(0)} \leftarrow \arg\min_{\beta \in \mathbb{R}^d} \|X\beta - y\|_2^2$
3: $\quad \tilde{X} \leftarrow [(X^T)_2; \ldots; (X^T)_d]$
4: $\quad$ Sample $i_1, \ldots, i_m \in [n]$ without replacement
5: $\quad \mathcal{H} \leftarrow \{\}$
6: $\quad$ **for** $j \in [m]$ **do**
7: $\quad\quad$ **for** $t \in L$ **do**
8: $\quad\quad\quad$ Add $(\tilde{X}_{i_j}, y_{i_j} + t)$ to $\mathcal{H}$
9: $\quad \mathcal{R} \leftarrow$ GENERATEREGIONS($\mathcal{H}$)
10: $\quad$ **for** $R \in \mathcal{R}$ **do**
11: $\quad\quad$ For each $i \in [n]$, compute

$$l_i = \inf_{\lambda \in R} \langle \tilde{X}_i, \lambda \rangle - y_i \qquad \text{and} \qquad r_i = \sup_{\lambda \in R} \langle \tilde{X}_i, \lambda \rangle - y_i.$$

12: $\quad\quad$ Set $K_+ = \{i \in [n] : l_i \geq 0\}$ and $K_- = \{i \in [n] : r_i \leq 0\} \setminus K_-$.
13: $\quad\quad$ Solve linear program

$$\hat{S}(R) \leftarrow \inf_{g \in \mathbb{R}^n, \lambda \in R} \left\{ \sum_{i \in K_+} \frac{g_i}{r_i} + \sum_{i \in K_-} \frac{g_i}{l_i} \;\middle|\; \begin{array}{ll} X^T g = X^T(\tilde{X}\lambda - y) & \\ 0 \leq g_i \leq \langle \tilde{X}_i, \lambda \rangle - y_i & \forall i \in K_+ \\ \langle \tilde{X}_i, \lambda \rangle - y_i \leq g_i \leq 0 & \forall i \in K_- \\ l_i \leq g_i \leq r_i & \forall i \in [n] \setminus (K_+ \cup K_-) \end{array} \right\}.$$

14: $\quad\quad$ For each $i \in K_+ \cup K_-$ let $Q_i = \min(l_i^2, r_i^2)$; for each $i \in [n] \setminus (K_+ \cup K_-)$ let $Q_i = 0$.
15: $\quad\quad$ Let $Q^{(1)} \leq \cdots \leq Q^{(n)}$ be the sorted numbers $Q_1, \ldots, Q_n$ and compute

$$\hat{N}(R) \leftarrow \sup \left\{ k \;\middle|\; \sum_{i=1}^{n-k} Q^{(i)} > \left\| X\beta^{(0)} - y \right\|_2^2 \right\}.$$

16: $\quad$ **return**

$$\min_{R \in \mathcal{R}} \max(\hat{S}(R), \hat{N}(R)).$$

---

