# OpenReview forum: "Provably Auditing Ordinary Least Squares in Low Dimensions"
_ICLR.cc/2023/Conference — ICLR 2023 poster_

### Official Review · Reviewer_BXKi · 2022-10-23

**Confidence:** 4
**Correctness:** 4
**Technical Novelty And Significance:** 4
**Empirical Novelty And Significance:** 4
**Recommendation:** 8

**Clarity, Quality, Novelty And Reproducibility:**

I appreciate the technical merits of the work.  The paper is well presented, and the algorithms are sufficiently explained at an intuitive level first in the main paper then formally described and analyzed in the appendix. The main results, to the best of my knowledge, are novel.

**Strength And Weaknesses:**

The paper is clearly strong at a technical level. The results are sound and complete. The upper bound for exact estimation, though somewhat unsatisfying in terms of run-time, is complemented by the faster approximation algorithm and the lower bound.

The author(s) provide an extensive set of experiments, which is great for a work that is theoretical by nature.

That said, I can see that the paper can be strengthened in a few ways.

1. Can any of these results by extended to regularized (ridge) regression?
2. Is it possible to identify certain distributional assumptions so that the run-time lower bound can be bypassed? It would be great to demonstrate poly-time algorithms even though it has be to under further conditions.
3. The prior work of Broderick et al., 2020 (https://arxiv.org/abs/2011.14999) considers a broader setting, where there's a general quantity of interest $\phi : \mathbb R^d → \mathbb R$ (mapping from the $d$-dimensional parameter $\beta$ to a real) and the algorithm is asked to check whether $\phi$ is stable wrt the dataset. The setting studied in this work is strictly more restricted, since it's only concerned with a single coordinate.  I am wondering if the results can be extended further.

**Summary Of The Paper:**

The paper provides algorithms for estimating stability of OLS to dropping a small fraction of samples. The theoretical results focus on low-dimensional regime and give algorithms with run-time exponential in $d$.

**Summary Of The Review:**

Given the strengths of the paper, I recommend accept.

---

> ### Author Response · Authors · 2022-11-10
> **Response**
>
> Thanks for reading our paper and appreciating our results! Questions (1) and (2) are super interesting directions for future research, but we don't have answers at the moment. As for (3), we remark that there is a black-box reduction to our problem from any *linear* quantity of interest, i.e. $\phi_t(\beta) = \langle \beta, t\rangle$ for a test vector $t$. The reduction is to first apply the transformation $X \mapsto XO$ for an appropriate orthonormal matrix $O$. For any weight matrix $W = \text{diag}(w)$, the weighted OLS solution is described by $\hat{\beta}(XO,y,w) = O^T (X^T W X)^{-1} X^T W y = O^T \hat{\beta}(X,y,w).$
> Stability of the first coordinate of $\hat{\beta}(XO,y,w)$ is therefore equivalent to stability of $\phi_{(O^T)_1}(\beta) = \langle \beta, (O^T)_1\rangle$ on the original dataset.
>
> We can also compute stability of $\phi(\beta) := \beta_1 - c$, i.e. how many samples need to be dropped to make the first coordinate at most $c$. The reduction is to first apply the transformation $y \mapsto y - c(X^T)_1$.
>
> However, the prior work of (Broderick et al., 2020) can also handle nonlinear functionals, and this is indeed a limitation of our results.

---

> > ### Comment · Reviewer_BXKi · 2022-11-16
> > **Thanks**
> >
> > Thank you for the response! I'd like to suggest adding this black-box reduction of linear quantity as a remark to the paper.
> >
> > I have no further comment and would recommend accept.

---

### Official Review · Reviewer_hNxp · 2022-10-24

**Confidence:** 2
**Correctness:** 3
**Technical Novelty And Significance:** 3
**Empirical Novelty And Significance:** 3
**Recommendation:** 8

**Clarity, Quality, Novelty And Reproducibility:**

The paper is of good quality and novelty, but is not perfectly clear.
Below, I ask some questions about intuitions in the proof sketches, since I am not convinced of correctness when I don't understand proof sketches. Afterwards, I list some mild typos or writing suggestions.

## Proof Sketch Questions

1. [Page 4, first paragraph] Appendix F.1 seems to give polynomially small mild anti-concentration, not exponentially small mild anti-concentration.  What's the relationship between Proposition F.1 and $(\varepsilon, e^{-\Omega(n)})$?
1. [Page 5, remark 3.1] Why do such $w'$ have to lie in a linear subspace of bounded codimension? How does this argument follow?
1. [Page 6, second-to-last paragraph] I don't see where the hyperplanes come in from. I can assume that the residuals are bounded, but how does this impose some sort of linear geometry? How does $\varepsilon$ appear in this picture?
1. [Page 6, paragraph after equation 2] This is altogether okay, since it seems to just be appealing to the prior work, but is there a clearer intuition you can give for what parts of this geometry go into appealing to the prior work? The sentence "By classical results..." seems to carry a crux of the proof, but I don't get what geometry it's even using.

## Minor suggested edits and typos

1. I'd recommend changing the language a bit from "finite-sample stability" to "fractional stability" and "integral stability" to "discrete stability" or "binary stability". _Finite-sample_ doesn't sound fractional, and _integral_ can sound continuous since integrals are continuous sums. _Finite-sample_ is definitely worse than _integral_ though.
1. Instead of the $\star$ operator, consider the standard notation for the [Hadamard product](https://en.wikipedia.org/wiki/Hadamard_product_(matrices)), or using a diagonal matrix to contain $w$, since that's pretty common in the weighted least square (and broader linear algebra) community.
1. WLS is a standard acronym for weighted least squares, so I'd recommend always saying either WLS or OLS, where the former has a weight vector and the latter does not have a weight vector.
1. [Page 3, paragraph after theorem 1.3, sentence that starts with "In practice"] Say "model" or "dataset" instead of "conclusion"
1. [Page 8, Isotropic Gaussian Data paragraph] Say "this is possible" instead of "this is the case".
1. [Page 8, last paragraph] "Just 27% of the data" -- isn't that pretty large? Why should we expect or especially want more than a quarter of the data to control just the sign of a single variable from OLS?
1. [Pages 8,9; the plots] The labels of the axes should be __much__ larger, so they're easily legible. Also, figure 2(b) is missing error bars.
1. [Page 9, All-features-pairs analysis] "and never performs much worse" -- does it ever perform any worse? It seems to lie entirely below the line $y=x$ on the plot. Would be good to clarify in the text.

**Strength And Weaknesses:**

I like this paper. I recommend it be published.

At a high level, it takes an intractable problem (the prior work's notion discrete of stability), keeps the intuitive underlying structure, and extends it to a fractional problem that's relatively tractable.

The paper is decently well written, with the high level idea of the algorithms being broadly clear, but some parts of the proof sketches being hard to understand. The results form a decently satisfying narrative: a terrible runtime without any assumptions, a lower bound to boot, a mild assumption to improve the terrible runtime with some approximation error, and a stronger assumption to get a potentially tractable runtime with some approximation error.

The results seem significant. The lower bound from the exponential time hypothesis makes it clear that this problem is hard, so fighting to get a runtime that's just polynomial in $n$ makes sense. Further, since verifying the stability of models is high-stakes in high-stakes applications, we really don't want to rely on heuristic algorithms. Provably correct algorithms are important. So, while the fastest runtime presented is pretty horrendous for even small $d$, it's nevertheless a good win.

The results certainly have weaknesses though, though I'm inclined to be lenient towards these weaknesses given the hardness of the problem. The key weaknesses are:
1. The exact algorithm runs in $n^{O(d^3)}$, the lower bound for exact algorithms is $n^{\Omega(d)}$, and the approximate algorithm with mild-anticoncentration assumption runs in $\approx n^{O(d)}$ time. In order to just match the lower bound, the paper both relaxes the exactness of the answer and assumes structure on the input.

    $\phantom{.}$

    It's not clear _where there's looseness_. Can an exact algorithm run faster, at $n^{O(d)}$? Does mild-anticoncentration suffice to beat the $n^{\Omega(d)}$ lower bound, just using a different algorithm? Does approximation error allow us to beat the $n^{\Omega(d)}$ lower bound? Is there a $n^{\Omega(d)}$ lower bound for approximate algorithms given inputs that satisfy mild-anticoncentration? There's some gap here that I don't really understand, nor have any intuition for.

    $\phantom{.}$

2. The proof sketches are a bit unclear. I can follow 90% of the intuition, but then some language about _demarcating regions_ or _connected component of varieties_ appears to resolve a conceptual technical block, and I don't really get how this work. I'm sure these are elaborated upon in the appendix, but that shouldn't be necessary. This hurts my confidence of correctness of the algorithms.

    $\phantom{.}$

3. The final approximation algorithm, with it's stronger assumption on the input, is similar to the _incoherence_ assumption made in the matrix completion literature, which intuitively says that sampling rows from a matrix uniformly-at-random suffices to preserve $\ell_2$ norm information (see [here for a formal paper](https://arxiv.org/pdf/1408.5099.pdf) and [here for a more approachable writeup](https://randnla.github.io/leverage-subspace-embedding/#uniform-sampling)). It's certainly weaker than a real incoherence assumption, since an $\varepsilon$-fraction of rows are allowed to violate incoherence, but we can still contrast the assumptions.

    $\phantom{.}$

    For an incoherent matrix, any uniform subsampling of $O(d \log d)$ rows preserves the least squares solution with high probability. So, we should intuitively find that such matrices have to be stable. Admittedly, this notion of "preserving" a solution is not obviously related to stability, so my intuition could be really betraying me here. At the same time though, it remains unclear how much of the weight of this result is owed to the algorithm, and how much is owed to the incoherence-like assumption. Either way, it's a nice result to have out there, but I think there's a nice discussion to be had about how this differs from incoherence.

4. Both assumptions on the input data, the mild-anticoncentration and the near-incoherence, read sorta like assumptions on stability. It's not clear if we should be assuming some kind of stability to


---
### Appendix: Formalizing the Incoherence Relationship
For the authors, I formalize what I see at the relationship between incoherence and the assumption used for the last algorithm of their paper. First the _leverage score_ of row $i$ of a matrix $\bar{X}$ is
$$
\tau_i[\bar{X}] = \max_{\beta} \frac{([\bar{X}\beta]_i)^2}{\|\|\bar{X}\beta\|\|_2^2}
$$
Then $\tau^* := \max_i \tau_i$ is the incoherence of the matrix. In other words,
$$
\forall \beta,\ \forall i\in[n],\ \frac{([\bar{X}\beta]_i)^2}{\|\|\bar{X}\beta\|\|_2^2} = \tau^*
$$
Assumption B in the paper, when $\varepsilon \rightarrow 0$, says that $\tau^* = \frac{\delta^2}{n}$, so that (intuitively) uniform $\tilde{O}(\delta^2 \log(d))$ rows is needed to preserve $\ell_2$ norms (assuming we pick $\delta$ to be as small as possible for $\bar{X}$).

**Summary Of The Paper:**

Given a matrix $X\in\mathbb{R}^{n \times d}$ and vector $y \in \mathbb{R}^{n}$, we are interested in understanding the _stability_ of the least squares problem $\min_{\beta} \|\|X \beta - y\|\|_2$. In the prior work before this paper, _stability_ is defined as the smallest number of rows of $X$ that need to be remove to make the (wlog) first entry of the least squares solution $\beta$ flip sign. This is computationally hard, and so the prior work relies on heuristic or brute-force algorithms.

This paper considers a fractional notion of stability, where we are able to reweigh each row of $X \beta - y$ by some $w_i \in [0,1]$, where $w_i=0$ means entirely removing the row and $w_i=1$ means keeping the row, and ask what is the smallest sum of $(1-w_i)$ needed to make the first entry of the least squares solution flip sign. The paper focuses on the computational hardness of this problem, and shows three key results:
1. Exactly computing fractional stability is possible in $n^{O(d^3)}$ time.
1. Exactly computing fractional stability requires $n^{\Omega(d)}$ time, by reduction to the Exponential Time Hypothesis.
1. Under a "mild anti-concentration" assumption that few rows of $[X \beta]$ are nearly linearly independent from other rows, a $\tilde{O}(n + \frac{d}{\varepsilon^2})^{d + O(1)}$ time algorithm approximates fractional stability to additive error
1. Under a stronger near-incoherence assumption, a $O((\frac{\sqrt d}{\varepsilon})^d \text{poly}(n))$ time algorithm approximates fractional stability to additive error

Compelling experimental results show that these algorithms can be practical on low-dimensional datasets, giving provable stability in potentially tractable runtime.

**Summary Of The Review:**

The setup and results are cool, but I don't completely understand the technical proof sketches for correctness. I understand a lot of it though, so I'm still confident enough to give the paper a pass.

---

> ### Author Response · Authors · 2022-11-10
> **Response (part 1)**
>
> We thank the reviewer for reading our paper and for thoughtful questions and suggestions. To address the reviewer's critiques/questions:
>
> 1. *(Re. tightening the landscape of assumptions vs computational tractability)* These are very interesting questions to which we unfortunately don't have great answers :). To give our limited intuition/speculations:
>
> > Is there an exact algorithm achieving $n^{O(d)}$?
>
> We believe that there ought to be, because Theorem 1.3 seems to come quite close: assuming $\text{poly}(n)$ bit complexity of $(X,y)$ (which is implicitly used even for Theorem 1.1, unless we are allowed exact arithmetic over the real numbers) and assuming that the optimal weight vector $w$ has $\text{poly}(n)$ bit complexity, there is a bound of $\text{poly}(n)$ on the bit complexity of an optimal $\lambda$ in Equation (3). But Assumption A essentially holds for $\beta \in \mathbb{R}^d$ with polynomial bit complexity, with $\epsilon = \exp(-\Omega(n))$ and $\delta = 0$ (there could be points contained exactly in a hyperplane, but this isn't an issue for the
> PartitionAndApprox algorithm; only points close-but-not-in a hyperplane cause issues). Hence, exact computation of stability seems potentially achievable in $n^{O(d)}$ time (modulo bit complexity issues, and lots of details in the above sketch that we haven't checked).
>
> > Does approximation error allow us to beat the $n^{\Omega(d)}$ lower bound?
>
> We are unsure which way to bet for this, because the hard instances in our current lower bound (via Maximum Feasible Subsystem) can be solved in fixed-parameter tractable time when we allow $\epsilon n$-additive error for constant $\epsilon$, and it's not clear how to construct hard instances if not via Maximum Feasible Subsystem.
>
> > Does mild anti-concentration suffice to beat the $n^{\Omega(d)}$ lower bound?
>
> This is a particularly exciting question because a positive answer seems plausible. Perhaps cleverly combining PartitionAndApprox and NetApprox could work (the former pays $n^{\Omega(d)}$ because it fundamentally has to partition the space of possible $\lambda$ by the signs of residuals of all data points, but the latter somehow avoids this). Unfortunately we were not able to prove such a result, and it seems that new algorithmic ideas may be necessary.
>
> 2. *(Re. jargon in proof sketches)*
>
> We'll try to clarify these parts of the overview. To be precise:
>
> > By classical results on connected components of varieties...
>
> This phrase was probably unnecessary and perhaps caused confusion. At this point of the overview, we're simply appealing to Theorem B.3 (due to Milnor and Renegar), which is closely related to important results about Betti numbers of algebraic varieties.
>
> > The regions could be demarcated by $O(...)$ hyperplanes
>
> This was informal language with the following formal meaning. For hyperplanes $(h_i(x))_i$, the regions ``demarcated'' by these hyperplanes are the closures of the connected components of $\mathbb{R}^d \setminus \bigcup_i \\\{x \in \mathbb{R}^d: h_i(x) = 0\\\}$. Each of these regions is a convex polytope; in particular, it's described by the linear inequalities $(\sigma_i h_i(x) \geq 0)$ for some sign vector $\sigma \in \\\{-1,1\\\}^d$.
>
> Hopefully this helps clarify matters? Please let us know if you have further questions about the proof sketches!

---

> ### Author Response · Authors · 2022-11-10
> **Response (part 2)**
>
> 3. *(Re. leverage scores and incoherence)*
>
> Thanks for pointing out the connection with leverage scores / incoherence / subsampling. Indeed, Assumption B seems very closely related to incoherence. However, it's not the case that incoherence implies large finite-sample stability. We would articulate the relation between these notions as follows: incoherence certainly may imply an *average-case* version of stability, where dropping samples uniformly at random preserves the regressor. In contrast, finite-sample stability is a worst-case metric where the dropped samples are chosen adversarially.
>
> These notions diverge even in simple settings. Consider multivariate Gaussian data, i.e. with independent samples $(X_i,y_i)_{i=1}^n$ generated as $X_i \sim N(0,\Sigma)$ and $y_i = \langle X_i, \beta^*\rangle + N(0,\sigma^2)$. Then we can make several observations:
>
> - Finite-sample stability roughly measures the signal-to-noise ratio (see e.g. Figure 2), because as the noise increases, we can make larger perturbations to the regressor by adversarially selecting samples where the noise correlates with the covariates in a particular direction. That is, as $\sigma/|\beta^*_1|$ becomes a larger and larger constant, the fraction $\epsilon$ of samples that need to be dropped to change the sign of the first coordinate of the regressor becomes a smaller and smaller constant.
>
> - In contrast, when subsampling $k$ data points, the perturbations in the regressor will be roughly on the order of $\sigma/\sqrt{k}$. So even if we randomly drop $9/10$ths of the samples, we cannot change the sign of the first coordinate of the regressor unless $\sigma/|\beta^*_1| = \Omega(\sqrt{n})$.
>
> - Assumption B is indeed satisfied with high probability, for an arbitrarily small constant $\epsilon > 0$ and $\delta := \epsilon/4$, regardless of the signal-to-noise ratio (Proposition F.2).
>
> 4. *(Re. are assumptions A and B assumptions on stability)*
>
> See response to (3). Both assumptions are satisfied by Gaussian data even when the signal-to-noise ratio is very low (and hence the finite-sample stability is very low). Additionally, note that experimentally our algorithms do succeed at diagnosing datasets with very low stability.

---

> ### Author Response · Authors · 2022-11-10
> **Response (part 3)**
>
> To address the proof sketch questions:
>
> 1. Yes, Proposition F.1 gives $(\epsilon, 1/\text{poly}(n))$ anti-concentration when $\sigma = 1/\text{poly}(n)$. However, note that this is \emph{better} than $(\epsilon, \exp(-\Omega(n)))$-anti-concentration: the former implies the latter. Proposition F.1 implies that even if $\sigma$ is taken to be $\exp(-n)$ (i.e. inverse-exponential smoothing), anti-concentration is still satisfied with parameters $(\epsilon, \exp(-\Omega(n)))$, which is good enough to get an $n^{O(d)}$-time algorithm.
>
> 2. Fix a feasible $(w,\lambda)$. Consider the set of $w'$ such that $(w,\lambda)$ is feasible and $\lVert w\rVert_1 = \lVert w'\rVert_1$. The latter is a linear constraint (since we're already restricted to $[0,1]^n$). The former is the constraint $X^T(w' \star (\tilde{X}\lambda - y)) = 0$, which consists of $d$ linear constraints. Thus, this set is $[0,1]^n \cap V$ for a subspace $V$ of codimension at most $d+1$.
>
> Now the claim follows from a greedy rounding argument. Suppose that $w$ has more than $d+1$ non-integer coordinates; let $S = \{i \in [n]: 0 < w_i < 1\}$. By dimension counting, there is a vector $\alpha \in \mathbb{R}^n$ which is (a) supported in $S$, and (b) lies in $V$. Then there is some $c \in \mathbb{R}$ such that $w' := w + c\alpha$ still lies in $[0,1]^n \cap V$ and has at least one more integer coordinate than $w$. Moreover, $(w',\lambda)$ is still feasible and we haven't changed the $\ell_1$ norm. So we repeat until the weight vector has at most $d+1$ nonzero coordinates.
>
> 3. (Second-to-last paragraph of page $6$) The main idea here is as follows. For any polytope $R$ (where each residual has constant sign on the region), the program $V_R$ (defined a paragraph earlier) has linear constraints, but the objective function is nonlinear. But suppose the following ``Property $(\dagger)$'' were true: for every $i \in [n]$, the numbers $\max_{\lambda \in R} \langle X_i, \lambda\rangle - y_i$ and $\min_{\lambda \in R} \langle X_i,\lambda\rangle - y_i$ not only have the same sign, but also are within a $(1 \pm \epsilon)$ multiplicative factor. Then the objective function can be multiplicatively approximated up to $(1 \pm \epsilon)$ by a linear function, where we replace $\sum_{i=1}^n g_i / (\langle X_i,\lambda \rangle - y_i)$ by $\sum_{i =1}^n g_i / (\min_{\lambda' \in R} \langle X_i,\lambda'\rangle - y_i)$. Note that the denominators are just numbers, so this objective is linear in the variables of the program.
>
> Of course, Property $(\dagger)$ does not hold for $\mathbb{R}^d$, so we need to partition $\mathbb{R}^d$ into polytopes which do satisfy $(\dagger)$, and solve the LP for each polytope separately. To do this, consider the set of hyperplanes $\langle X_i,\lambda\rangle - y_i = \pm (1+\epsilon)^k \cdot \delta M /\sqrt{n}$ as we vary $k$ from $0$ to $\log_{1+\epsilon}(\sqrt{n}/\delta)$. Also consider the set of hyperplanes $\langle X_i,\lambda\rangle - y_i = 0$. All of the polytopes ``demarcated'' by these hyperplanes (i.e. the connected components of the complement of the union of the hyperplanes) satisfy Property $(\dagger)$. Since we have a bound on the number of hyperplanes, we can bound the number of polytopes which the algorithm needs to consider.
>
> All of this was under the ``wishful thinking'' that all residuals are always between $\delta M/\sqrt{n}$ and $M$. Obviously, this cannot be true. However, under anti-concentration, it's nearly true: for any $\lambda$, at most $\epsilon n$ residuals are outside those bounds. As a result, we can prove that the whole argument works with an additional additive error of $O(\epsilon n)$.
>
> 4. (Paragraph after Equation 2) The question at hand is how to bound the number of achievable ``relative orderings'' of the numbers $(\langle X_i,\lambda\rangle - y_i)\langle X_i,u\rangle$ as we vary $(\lambda, u) \in \mathbb{R}^{2d-1}$. But this is bounded by the number of connected components of $\mathbb{R}^{2d-1} \setminus \cup_{i,j} \{(\lambda,u): f_{i,j}(\lambda,u) = 0\}$ where $f_{i,j}$ is the quadratic polynomial $f_{i,j}(\lambda,u) = (\langle X_i,\lambda\rangle - y_i)\langle X_i,u\rangle - (\langle X_j,\lambda\rangle - y_j)\langle X_j,u\rangle$. The reason is that in every connected component, for any $i,j$, the polynomial $f_{i,j}$ has constant sign.
>
> So the question is now, how many connected components can the zero sets of $n^2$ quadratic polynomials in $2d-1$ variables induce? This is a non-trivial question and we appeal to results by Milnor and Renegar which show that the answer is $n^{O(d)}$ (rather than $\exp(O(n))$. The simple geometric intuition is that this is just a higher-degree generalization of an elementary bound for linear functions: in $\mathbb{R}^2$, $n$ lines can divide the plane into at most $O(n^2)$ regions, and in $d$ dimensions, a simple induction shows that $n$ hyperplanes can produce at most $O(n^{d+1})$ regions.

---

> ### Author Response · Authors · 2022-11-10
> **Response (part 4)**
>
> Thanks for the suggested edits. A few comments in response:
>
> 6. We are contrasting $27\\\%$ with the (far weaker) bound of $90\\\%$ produced by the greedy heuristic
>
> 7. We discuss error bars in Appendix I.3; there are no error bars in I.3 because we only had one run of that experiment due to computational constraints.
>
> 8. It's a little difficult to see on the plot, but there are a few points slightly above the diagonal (i.e. where it performs slightly worse than the greedy heuristic); we can add a clarification in-text.

---

### Official Review · Reviewer_LbgB · 2022-10-25

**Confidence:** 3
**Correctness:** 3
**Technical Novelty And Significance:** 3
**Empirical Novelty And Significance:** 2
**Recommendation:** 6

**Clarity, Quality, Novelty And Reproducibility:**

- Clarity: the presentation of the paper is mostly clear. Question: what is the significance of zeroing out the first coordinate of the coefficient vector in the stability definition?
- Quality: theoretical analysis is provided to justify the quality of the approach.
- Novelty: the proposed approximation algorithm is novel; however, generalization of the method to other stability metrics are not straightforward.


**Strength And Weaknesses:**

[Strengths]
- Evaluating the stability of OLS could be useful for robust regression and model interpretation.
- The paper presents polynomial-time algorithms for estimating a particular stability metric introduced in (Broderick et al., 2020). Theoretical guarantees are also provides for the approximation quality.
- The authors illustrated the utility of the method on Boston Housing Data.

[Weaknesses]
- The proposed method seems to be limited to only linear models/OLS problems. Generalizations to non-linear models or more advanced models are not obvious.
- The proposed method targets one particular stability metric (Broderick et al., 2020). This metric might be sensitive to noise in the data or resampling of the data. There are widely-studied candidate stability metrics such as the Cook's distance in regression analysis that could afford cheaper computations.


**Summary Of The Paper:**

The paper presents a fast algorithm to estimate a stability metric of ordinary least squares problems (OLS). The stability metric was introduced as
the minimum number of samples that need to be removed so that rerunning the analysis overturns the conclusion. Naive computation of the stability metric is computationally prohibitive, whereas the proposed algorithm can efficiently obtain an approximation in the low-dimensional regime.


**Summary Of The Review:**

The authors propose an efficient approximation algorithm for computing the stability metric introduced in (Broderick et al., 2020). Theoretical analysis is provided to justify the robustness of the algorithm.

---

> ### Author Response · Authors · 2022-11-10
> **Response**
>
> Thanks for reading our paper and for the questions and comments!
>
> > The proposed method seems to be limited to only linear models/OLS problems.
>
> Indeed, we started with OLS because it's the most fundamental and widely-used statistical estimator, but extending our results to more general estimators (e.g. the generalized method of moments estimator, or ridge regression as pointed out by Reviewer BXKi) is an interesting (if perhaps difficult) direction for future work.
>
> > The proposed method targets one particular stability metric (Broderick et al., 2020). This metric might be sensitive to noise in the data or resampling of the data.
>
> This is true, which is why we do not view this metric as the ``ultimate'' stability metric, but rather as one metric to be used in conjunction with complementary metrics \--- for example, a metric for stability to noise/perturbations of the data. Finite-sample stability does not supercede such a metric (e.g. it's easy to imagine a dataset where removing large fractions of the data can't change the regressor, but slightly perturbing the data points can cause a drastic change) but we would also like to emphasize that neither does a perturbation-stability metric supercede finite-sample stability. For example, a heterogeneous dataset with a small subset that drives the regressor in a particular direction would have very low finite-sample stability, but this instability would not be detected by perturbation-stability metrics.
>
> > There are widely-studied candidate stability metrics such as the Cook's distance in regression analysis that could afford cheaper computations.
>
> As always, there is a tradeoff curve between computation time and utility. As we discuss in Appendix A (``Further Related Work''), Cook's distance and most other widely-studied metrics in regression analysis / sensitivity analysis are fundamentally *local* metrics, because they measure the influence of an individual data point rather than groups. Such metrics are computationally cheap but sacrifice utility when they are used as proxies for group stability (a *global* metric which is arguably a more fundamental notion in large datasets).
>
> The high-level goal of this work was to seek ``higher-utility'' stability estimates (i.e. estimates for group stability with provable guarantees), which have been largely ignored in the literature on regression analysis, while retaining computational tractability for a broad class of datasets (in our case, low-dimensional datasets).
>
> > What is the significance of zeroing out the first coordinate of the coefficient vector in the stability definition?
>
> In practical applications of OLS, the sign of a particular coordinate of the regressor indicates whether we estimate that the corresponding covariate has a positive or negative effect on the response (after controlling for the other covariates): e.g., an economist might be interested in understanding whether education has a positive or negative effect on wages after controlling for parents' wages. If a coordinate can be easily zeroed out, it means that the estimated effect (positive or negative) of the original regression was not very robust.
>
> Obviously there is nothing special about the first coordinate, and the same algorithms apply to any particular coordinate of interest. Moreover, as we mention in our response to Reviewer BXKi, our definition is essentially equivalent (by black-box reduction) to measuring stability of the sign of any linear functional $\phi: \mathbb{R}^d \to \mathbb{R}$ of the regressor.

---

### Official Review · Reviewer_Hpqm · 2022-10-25

**Confidence:** 4
**Correctness:** 4
**Technical Novelty And Significance:** 3
**Empirical Novelty And Significance:** 3
**Recommendation:** 8

**Clarity, Quality, Novelty And Reproducibility:**

The paper is clearly written, and provide good insight into studying stability of optimization problem.

**Strength And Weaknesses:**

Unlike the previous results using influence functions to approximate the local stability, the paper study the exact problem--which means it is able to identify perturbation of the solution beyond local regime. The theory and experiments are sound. However, even with the relaxed approximate algorithm, the computation cost seems still very big.

**Summary Of The Paper:**

The paper studies the stability of linear regression problem by defining it as the minimum samples (in $\ell_1$-norm) to be removed to zero out the first coordinate. The paper provides a $O(n^{d^3})$ algorithm to exactly solve the problem, and also provide a lower bound stating that the computation complexity cannot be $O(n^{o(d)})$. Further, approximate algorithms are provided and experiments are carried out on real dataset to identify stability.

**Summary Of The Review:**

The paper is well written and organized. The problem is clearly formulated, and the authors tackle the exact stability problem instead of using local approximations from influence functions, which is novel and interesting to me. By exactly auditing linear regression, the authors provide theoretical guarantees for such identification and the algorithm works well on real datasets. Although the complexity of the algorithm seems to suffer from curse of dimensionality, but it is a good starting point for study stability problem in its exact form. I think this is a good paper and would like to accept.

---

> ### Author Response · Authors · 2022-11-10
> **Response**
>
> Thanks for reading our paper and appreciating our results! We entirely agree that the curse of dimensionality is a limitation of our current results, and evading it under stronger assumptions is a super interesting direction for future work.

---

### Decision · Program_Chairs · 2023-01-20

**Decision:**

Accept: poster

**Justification For Why Not Higher Score:**

The proposed method is limited to linear models/OLS problems.
It targets only one particular stability metric (Broderick et al., 2020) that might be sensitive to noise in the data or resampling of the data.

**Justification For Why Not Lower Score:**

Evaluating the stability of OLS could be useful for robust regression and model interpretation.
The paper presents polynomial-time algorithms for estimating a particular stability metric introduced in (Broderick et al., 2020). Theoretical guarantees are also provided for the approximation quality.

**Metareview: Summary, Strengths And Weaknesses:**

The paper studies the stability of linear regression problem by defining it as the minimum samples to be removed to zero out the first coordinate. The paper provides an algorithm to exactly solve the problem (an approximation can be obtained in the low-dimensional regime) and also provides a lower bound stating that the computation complexity cannot improved much.

**Note From Pc:**

if the above contains the word "oral" or "spotlight" please see: "oral" presentation means -> notable-top-5% and "spotlight" means -> notable-top-25%. As stated in our emails, we are disassociating presentation type from AC recommendations